



# A novel method for objective identification of 3-D potential vorticity anomalies

Christoph Fischer[1,2], Andreas H. Fink[2], Elmar Schömer[1], Roderick van der Linden[2], Michael Maier-Gerber[2], Marc Rautenhaus[3], and Michael Riemer[4]

[1]Institute of Computer Science, Johannes-Gutenberg University, Mainz, Germany
[2]Institute of Meteorology and Climate Research, Karlsruhe Institute of Technology, Karlsruhe, Germany
[3]Regional Computing Centre, Visual Data Analysis Group, University of Hamburg, Hamburg, Germany
[4]Institute for Atmospheric Physics, Johannes-Gutenberg University, Mainz, Germany

**Correspondence:** Christoph Fischer (christoph.fischer@uni-mainz.de)

**Abstract.** Potential vorticity (PV) analysis plays a central role in studying atmospheric dynamics and in particular in studying the life cycle of weather systems. The three-dimensional (3-D) structure and temporal evolution of the associated PV anomalies, however, are not yet fully understood. An automated technique to objectively identify 3-D PV anomalies can help to shed light on 3-D atmospheric dynamics in specific case studies, as well as facilitate statistical evaluations within climatological

studies. Such a technique to identify PV anomalies fully in 3-D, however, does not yet exist. This study presents a novel algorithm for the objective identification of PV anomalies in gridded data, as commonly output by numerical simulation models. The algorithm is inspired by morphological image processing techniques and can be applied to both two-dimensional (2-D) and 3-D fields on vertically isentropic levels. The method maps input data to a horizontally stereographic projection and relies on an efficient computation of horizontal distances within the projected field. Candidates for PV anomaly features

are filtered according to heuristic criteria, and feature description vectors are obtained for further analysis. The generated feature descriptions are well suited for subsequent case studies of 3-D atmospheric dynamics as represented by the underlying numerical simulation, or for generation of climatologies of feature characteristics. We evaluate our approach by comparison with an existing 2-D technique, and demonstrate the full 3-D perspective by means of a case study of an extreme precipitation event that was dynamically linked to a prominent subtropical PV anomaly. The case study demonstrates variations in the 3-D

structure of the detected PV anomalies that would not have been captured by a 2-D method. We discuss further advantages of using a 3-D approach, including elimination of temporal inconsistencies in the detected features due to 3-D structural variation, and elimination of the need to manually select a specific isentropic level on which the anomalies are assumed to be best captured. The method is made available as open-source for straightforward use by the atmospheric community.

## 1 Introduction

Weather systems and extreme weather events result from nontrivial three-dimensional (3-D) interactions in the atmosphere. The potential vorticity (PV) perspective on atmospheric dynamics provides an often-used conceptual framework to understand such interactions (as reviewed by Hoskins et al., 1985). A key component of this framework is that PV is materially conserved





in the absence of nonconservative processes, a reasonable approximation for large-scale atmospheric flow. When considered on isentropic levels, the temporal evolution of PV at a given location is hence governed by advection only, which typically yields a rather smooth PV evolution. Of particular importance to the PV perspective is the tropopause, as it separates air masses with typically low PV values in the troposphere from typically high-PV air in the stratosphere[1]. A strong PV gradient across the tropopause has motivated the definition of the so-called dynamical tropopause as an upper-level PV isosurface in the literature. Typically used PV values range between 1.5 - 4 PVU, with PV = 2 PVU being an often-used standard value (e.g., Morgan and Nielsen-Gammon, 1998).

The tropopause slopes from higher isentropic levels in the tropics towards lower isentropic levels at the poles. On a given isentropic level, poleward excursions of the tropopause denote an anomaly of low PV, whereas equatorward excursions denote an anomaly of high PV. A relatively smooth meridional undulation of the tropopause signifies quasi-linear Rossby wave dynamics. In the context of extreme weather, however, much attention has been given to highly non-linear PV features, in particular zonally narrow and meridionally extended "tongues" of high PV that intrude equatorwards (e.g., Massacand et al., 1998), known in the literature as PV streamers (PVSs). These elongated anomalies can have complex life cycles and behavior: They are often linked to Rossby wave breaking (RWB) events (e.g., McIntyre and Palmer, 1983; Thorncroft et al., 1993), interact with surface cyclones (e.g., Galarneau Jr et al., 2015; Bentley et al., 2017; Maier-Gerber et al., 2019), and may detach from the main stratospheric high-PV reservoir on a given isentropic level, forming so-called PV cutoffs. These cutoffs can diabatically decay, be reabsorbed by the main reservoir, undergo further dynamical processes, and can be associated with extreme weather events (Portmann et al., 2021).

In the present study, we consider the objective identification PV structures. Objective identification of atmospheric features from numerical simulation output has proven beneficial for a number of applications in both atmospheric research and operational meteorology. Typical uses include statistical analysis such as the computation of climatologies of feature occurrence (e.g., Limbach et al., 2012; Dawe and Austin, 2012) and operational weather forecasting purposes (e.g. Hewson and Titley, 2010) including ensemble forecast analysis (e.g. Hewson and Titley, 2010; Rautenhaus et al., 2015a). Recently, combination of state-of-the-art interactive 3-D visualization techniques (Rautenhaus et al., 2018, provide a comprehensive survey) with 3-D detection of atmospheric features opened the door for comprehensive case studies of 3-D atmospheric dynamics (e.g., Rautenhaus et al., 2015a; Kern et al., 2018, 2019; Bader et al., 2019). With respect to objective identification of PV structures, Papin et al. (2020) recently provided an overview of identification techniques for PVSs. They classified the methods into techniques based on the reversal of the meridional PV gradient, and techniques based on distance thresholds along a 2-PVU contour. For example, a 2-D identification technique that has found wide acceptance and use, and may thus effectively serve as a state-of-the-art benchmark technique, has been introduced by Wernli and Sprenger (2007). This technique is based on a scale analysis, according to which the 2-PVU contour of a PVS contains points that are far apart in the direction along the contour, yet relatively close together regarding their great circle distance. Wernli and Sprenger (2007) applied this technique to generate a climatology from ERA-15 reanalysis data, which Kunz et al. (2015) later extended to a 33 year time period.

---

[1]PV is typically positive on the northern hemisphere and negative on the southern hemisphere. Low PV air here refers to low absolute values and high PV air to high absolute values. For notational convenience we consider the northern hemispheric situation only.





All techniques discussed by Papin et al. (2020) operate on 2-D data only. Furthermore, gradient-reversal techniques only consider a subset of PVSs (often specifically focusing on RWB events), which are sensitive to small changes in the input data. To comprehensively study the atmospheric dynamics associated with a weather event of interest, however, it is often important to consider the full 3-D structure of the atmosphere, including related PV features. For instance, Bithell et al. (1999) put effort into analyzing and visualizing the 3-D structure of atmospheric PV. They showed a wide spectrum of 3-D PV anomalies (PVAs, see below) that are difficult to analyze (or even to track over time) in 2-D vertical or horizontal (isentropic) cross-sections alone. They further demonstrated that the evolution of wave-related features along the tropopause may cover an extensive vertical range that is difficult to describe in the framework of a 2-D analysis. Therefore, Bithell et al. concluded that the 3-D perspective is a useful tool for the comprehensive study of the evolution of the dynamical tropopause and related weather systems.

We note that in 2-D analysis, the concepts of PVSs and PV cutoffs have been widely established. They are clearly defined and widely used. However, a 3-D anomaly stretching over multiple isentropes can exhibit both types, PVS and PV cutoff, depending on the considered level. Therefore, this 2-D classification is challenging for evaluations, and especially does not hold for an analysis of 3-D features. In this study and for 3-D analysis, we hence name these features PV anomalies (PVAs). These anomalies can exhibit characteristics of both streamers and cutoffs when considered on a single isentropic level only. 3-D cutoffs, i.e. fully isolated areas of stratospheric air surrounded by tropospheric air, are only rarely a result of detachment from the main reservoir and mostly appear in the lower troposphere due to latent heat release (e.g., Bennetts and Hoskins, 1979).

An identification of 3-D PV structures has previously been performed for subsets of anomalies only, although the importance of their vertical structure has been demonstrated in the literature (e.g. Portmann et al., 2021; Lamarque and Hess, 1994). For example, Škerlak et al. (2015) investigated the 3-D structure of one subset of PVAs, namely tropopause folds and their link to extreme weather events. These folds are defined as structures with multiple tropopause crossings in the vertical direction. Portmann et al. (2018, 2021) focused on cutoffs that are identified independently on multiple isentropic levels and are subsequently stacked vertically using an overlap and proximity heuristic. Regarding a 3-D field, cutoffs can be thought of as stalactites attached to the stratospheric PV reservoir. Portmann et al. (2021) found that cutoffs can be potentially complex 3-D features, which can intensify on higher levels while decaying on lower levels, resulting in a vertical (cross-isentropic) displacement of these features. Therefore, tracking PVSs on a chosen 2-D isentropic level is not always sufficient to analyze an event during its entire life cycle. As investigated by Bithell et al. (1999), stalactites and tropopause folds are only a subset of possible anomaly types in a 3-D PV field. They classified the complex structures they encountered as *tubes*, *spirals*, *stalactites*, *folds*, among others, demonstrating complex scenarios that are not discernible within a 2-D perspective. Existing approaches used in 2-D identification lack the concepts of 3-D cohesiveness and information regarding extent, genesis and evolution of these structures. For instance, using the complete 3-D field offers the possibility to consider long time periods without the need to adjust a selected isentropic level to changes occurring between different seasons.

In summary, while many previous studies contributed to identifying different aspects of PV features on single isentropic levels, the objective identification of fully 3-D PV structures is still an open challenge and motivates our work. We expect that





a 3-D identification method will yield the opportunity for novel analyses, including new aspects within climatological studies and comprehensive case studies of 3-D atmospheric dynamics. In the article at hand, we hence present a novel algorithm for the identification of PV anomalies, which can be applied not only to 2-D, but also to 3-D fields. In the latter case, the algorithm operates on the complete 3-D fields instead of on individual isentropic levels. To the best of our knowledge, this is

the first approach aiming for a strategy that identifies the full spectrum of PVAs in 3-D. We use image processing techniques for the identification, more specifically numerical solutions that base on morphological operators. These operators, which originate from analytical geometry, have been adapted to suit the requirements of a meteorological application. This adaptation mainly revolves around using physical units to ensure interpretability, while considering the properties of the used projection. Parameters of the identification are adjustable and the algorithm is usable on different resolutions and projections. A low-

dimensional and human-readable feature vector is computed for each identified PVA consisting of quantifiable measures, e.g., centroid, intensity or best-fit geometry. This memory-efficient feature representation can serve as a basis for statistical analyses or as potentially relevant predictors. The identified 3-D PVA features are also well suited for studying atmospheric dynamics by means of interactive 3-D visual analysis (IVA; cf. Rautenhaus et al., 2018). For example, reducing atmospheric processes of importance to concise visual depictions facilitates combination of multiple aspects of atmospheric dynamics in

a comprehensive 3-D display well suited for rapid exploration of the considered numerical simulation data (e.g., Rautenhaus et al., 2015a; Kern et al., 2019). Here, we demonstrate such analysis by incorporating the identified PVA features into the 3-D meteorological visualization framework "Met.3D" (Rautenhaus et al., 2015b), which we use to shed light on the 3-D structure of PVAs encountered during an extreme precipitation event previously investigated by Van der Linden et al. (2017).

This article is structured as follows. Section 2 introduces the basic principle of the algorithm. In Sect. 3, a distance measure

required for the method is introduced. The identification algorithm is described for 2-D data in Sect. 4, and compared to the method of Wernli and Sprenger (2007) in Sect. 5. Section 6 introduces the generalized 3-D algorithm, which we apply in Sect. 7 for a case study, discussing the potential of 3-D PVA analysis. The paper is concluded in Sect. 8.

## 2   Basic principle of the algorithm

The identification algorithm is motivated by so-called morphological image processing techniques. These operations, often

used on binary data, process images based on the characteristics of their shape (Dougherty, 2020). Structuring elements (mostly matrices used as masks) are used to decide on how pixels in the image are altered based on their surroundings. Morphological operations are often used for morphological noise suppression or shape analysis. In the following, we first consider the 2-D case to illustrate our approach. Afterwards the method will be expanded to 3-D. To make use of morphological techniques, anomalies in the PV field (e.g., intrusions, cutoffs etc.) can be thought of as morphological "noise". A "noise-free" PV field

then resembles an idealized state of the atmosphere, where the height of the tropopause is symmetrical around the poles.

Fig. 1a-c shows the process of a so-called morphological opening applied to a binary image, which could resemble a PVS. An opening is defined as an erosion followed by a dilation of the data. For the erosion (Fig. 1a-b), a structuring element (orange) is used as mask. This mask is shifted over the binary input image. Each pixel $(x, y)$ is kept as part of the domain only

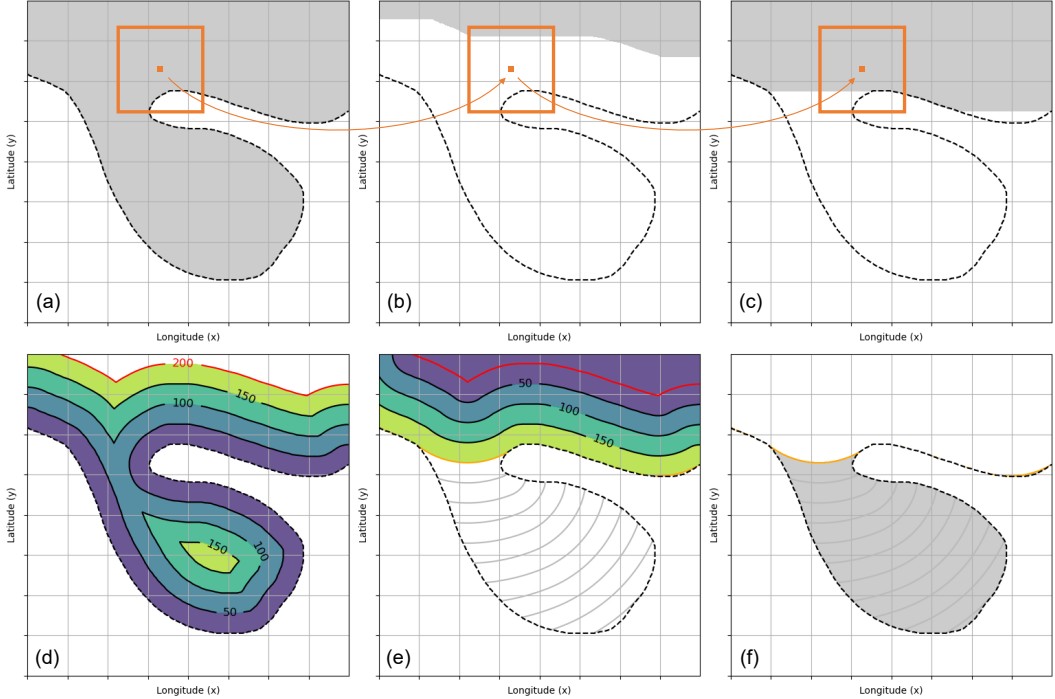

**Figure 1.** A sketch of our identification strategy applied to a PVS-like structure. (a-c): The process of a morphological opening (erosion followed by dilation) is applied to the gray structure using a square mask (orange). This removes thin structures depending on the mask's size. (d-f): Our adapted strategy, which uses distances instead of a binary mask. First, the distance emerging from the outer boundary is computed. Then, starting at a set distance (red line), the same distance is added back, respecting the boundaries of the domain. Areas not reconstructed are identified as anomalies.

if all pixels in the structuring element surrounding $(x, y)$ are also part of this domain. Thus, the erosion removes all areas along

edges depending on the size of the mask. As seen in the figure, it especially destroys elongated and filament-like structures.

Following the erosion, the dilation step (Fig. 1b-c) reconstructs areas of the domain. Using the same structuring element, a pixel $(x, y)$ is added back to the domain if any of the surrounding pixels defined by the structuring element is part of the domain. As seen in Fig. 1c, this reconstructs the general shape of the original object, but filament-like structures and other forms of noise are not reconstructed. These non-reconstructed areas can be extracted and interpreted as PVSs.

This approach uses an abstract concept of binary masks in discrete environments, and care is required to maintain interpretability and to design the algorithm with meaningful parameters. Fig. 1d-e shows an adaptation using Euclidean distances instead of a mask. Masks are typically square, and results based on these masks would give a distorted distance measure. The use of Euclidean distances in this Cartesian field results in an uniform effect along all directions. The adapted strategy works as followed: First, we measure distances emerging from the boundary (e.g., tropopause) into the domain, as seen in Fig. 1d. Then,

we choose a distance (e.g., the red line in Fig. 1d), and only keep areas as part of the domain that are further away from the





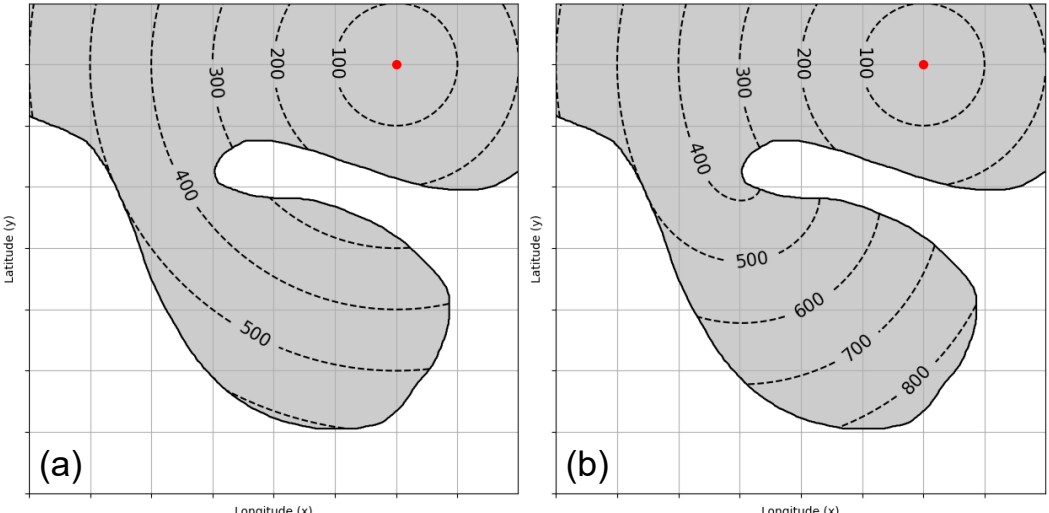

**Figure 2.** A visual example (same PVS-like structure as in Fig. 1) showing the shortest distance measure required for the algorithm. (a): Starting at the red dot, the direct spherical distance is shown, which can be calculated easily. (b): The distance following a well defined domain (e.g., PV field) is computed using a Fast marching method (FFM) as required by our strategy, see Sect. 3.2 for details.

boundary than the given parameter. A comparable parameter for the upper part of the figure would be the size of the structuring element. Then, starting from the newly defined boundary (red line), distances are measured outwards. Again, the result is at the given parameter (orange line in Fig. 1e-f). Areas further away than the value of the parameter are identified as anomalies.

Using this concept in real-world environments unfortunately is non-trivial. On the one hand, the distance measure required
must follow the domain instead of using a direct spherical distance, as seen in Fig. 2. Most algorithms that satisfy this require-
ment suffer from metrical errors induced by the discrete nature of algorithms. Furthermore, the real world closely resembles
a sphere and cannot be approximated using a Cartesian grid. Instead we have to consider the distortions of the projection our
algorithm is working in. The next section describes how distances following the field (as in Fig. 2b) are computed.

## 3 Solving the distance measure problem

### 3.1 Stereographic projection

For the identification algorithm described in this study we choose a pole-centered stereographic projection (Fig. 3). Its point
of projection is on the surface of the sphere opposite to the tangent plane where the projection is created. Thus, we can use the
South Pole as point of projection to have the view centered around the North Pole. This creates a singularity at the south pole
while minimizing the distortions in the northern hemisphere (Snyder, 1987).
We choose this type of projection due to several advantages:




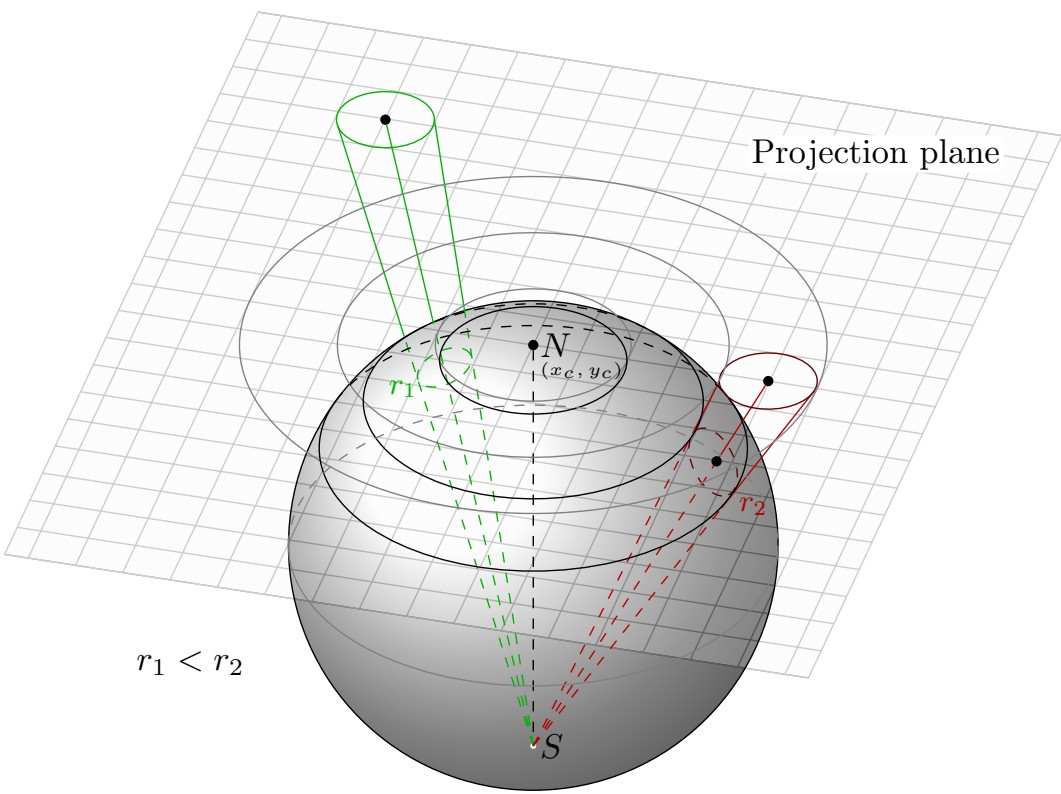

**Figure 3.** Depiction of the stereographic projection used in our identification strategy. This figure shows that circles on the sphere are mapped to circles on the projection plane. Therefore, this projection is conformal, a vital property for this strategy. Note that despite having the same shape, the sizes of the circles on the sphere vary by distance from the projection center.

1. The PV field encircles the earth and is centered around the poles. Using the pole as center of projection leads to a clear depiction of the features to be detected.

2. Contrary to equirectangular projections, this stereographic projection avoids singularities at the North Pole. PV intrusions over poles can occur, and their analysis is highly difficult in projections where a singularity is involved. Additionally, the stereographic projection solves issues handling the antimeridian boundary.

3. A stereographic projection is conformal, and the only known true perspective projection with this property (Snyder, 1987). A conformal projection preserves angles, thus, two lines on a sphere will intersect in the same angle as in their projection. As a result, infinitesimal shapes on a sphere are conserved in this projection. All Tissot's indicatrices are circles, therefore perfect circles on the sphere will always map to perfect circles in the projection, generally of a different area (see Fig. 3). This property will be useful to efficiently compute distances along the earth's surface regarding this projection later on. Note that map projections cannot be both area-preserving and conformal at the same time (Pearson, 1990).





The main drawback of this strategy is the introduction of a singularity at the projection center, i.e. the opposing pole. To avoid this problem while mapping global data, we create one stereographic projection centered around the North Pole and one centered around the South Pole, both extending towards the equator. The singularity at the pole shifts to a boundary at the equator. The sign of PV changes at the equator, and therefore it is questionable to identify PVAs over this boundary anyways.

## 3.2 Computing distances within the projection

As introduced in the previous section, our novel algorithm requires a way to calculate distances within the projection following a given field, as illustrated in Fig. 2. Here, we can use the conformal property of the projection. Since circles on the surface of a sphere are projected to circles in the stereographic projection, we can assign a distance field $\Delta s(x, y)$ for all points, resp. pixels $(x, y)$, in the projected field. Each pixel in the field $\Delta s$ contains the radius of a circle on the sphere in km that projects onto a circle of radius 1 pixel around $(x, y)$ in the projection. For example in Fig. 3, the center of green and red unit circle on the projection plane are assigned values $\Delta s(x, y) = r_i$, where $r_i$ is the radius of the corresponding circle on the sphere. Thus, based on the conformal properties, this distance can be expressed as a scalar value. The scalar field $\Delta s(x, y)$ then can be calculated using just the distortion factor at every point. For a sphere, according to Snyder (1987), the distortion factor $h$ at a point $(x, y)$ in a stereographic projection can be calculated by

$$h = \cos^{-2}\left(\frac{\theta(x, y)}{2}\right), \tag{1}$$

where $\theta(x, y)$ denotes the angle from the center of the projection at $(x, y)$. $h$ denotes the scale factor in all directions. For example, projecting the northern hemisphere with the North Pole as projection center yields a factor of 1 at the pole ($\theta = 0$) as expected, and a factor of 2 at the equator ($\theta = \frac{\pi}{2}$). Towards the South Pole, $h$ becomes infinite. The distance field $\Delta s(x, y)$ can simply be calculated by the pixel radius in km at the projection center $(x_c, y_c)$ (e.g., the pole, see Fig. 3), divided by the distortion factor:

$$\Delta s(x, y) = \frac{\Delta s(x_c, y_c)}{h}. \tag{2}$$

$\Delta s(x_c, y_c)$ itself can be calculated from the latitude and longitude positions of the pole and its surrounding pixels.

As next step, we define the distance map $u(x)$ containing the minimal distance from a well defined boundary $\Gamma$ to all $x$, where the distance $u(\Gamma)$ from the boundary to the boundary is zero. This problem is similar to the computation of signed distance functions or solving path planning problems. Popular algorithms to generate this field $u$ starting at $\Gamma$ contain variations of the Dijkstra algorithm (Dijkstra et al., 1959). Starting at the boundary, this construction is performed by evolving a level set to the destination based on a distance map (Petres et al., 2005).

However, most of these approaches suffer from metrication errors. They construct the path on segments parallel to the grid dimensions. Directions invariant to the grid orientation will always lead to errors induced by the $L_1$ norm (see Cohen and Kimmel, 1997). To get consistent results for the identification independent of the location of a PVS, these inaccuracies should be avoided.





This shortest path problem can also be formulated as path space integral. We search from all possible paths $P_{\Gamma,x}$ between
the boundary $\Gamma$ and point $x$ the shortest one respective to the cost function:

$$u(x) = \min_{P_{\Gamma,x}} \int \Delta s(z) dz. \qquad (3)$$

Generally, this formulation can be used with arbitrary cost functions, but replacing it with our distance field $\Delta s$ yields a cost
equal to the length of the path. This is a result of using a distance map which contains unit pixel distances in the projection.
For uniform $\Delta s$, i.e., a field without distortions, the formulation collapses to a simple euclidean distance expression.

Cohen and Kimmel (1997) explored that this path formulation satisfies the Eikonal equation

$$||\nabla u(x)|| = \Delta s(x). \qquad (4)$$

This first-order nonlinear differential equation is used widely in scientific applications, mainly in wave propagation problems
as a front propagation approach. In these cases, the cost function on the right-hand side is typically expressed as a propagation
velocity at every point in the domain. To our knowledge, we are the first to formulate this equation in a sense to compensate
for distorted distances in a projection instead, leading to a formulation for distances from a given boundary for conformal
projections. Intuitively, the gradient in the distance field $u$ is anti-proportional to the distortion of the distance field.

Solving the above differential equation yields a far more accurate result than graph based approaches on a discrete field,
resolving metrication errors. There are approaches solving the Eikonal equation, mainly the *Fast Marching Method* (FMM)
introduced by Sethian (1996). Starting at known distances in the distance field, i.e. at the boundary where the distance is zero,
the algorithm takes advantage of the fact that this field can iteratively be built from the boundary outward since the cost function
$\Delta s$ is strictly positive. We refer the interested reader to Petres et al. (2005) for a more in-depth description and evaluation about
this algorithm.

## 4 Identification technique in 2-D

Based on the above introduced strategies, we outline our identification technique for 2-D PVSs on a given isentropic level.
Figure 4 illustrates the individual steps; pseudo-code of the algorithm is shown in Algorithm 1.

### 4.1 Data

For the 2-D analysis, we use isentropic levels from both the ERA-5 reanalysis (Hersbach et al., 2020) data interpolated to
$0.5° \times 0.5°$ in both latitude and longitude, and data from the Subseasonal-to-seasonal (S2S) prediction database (Vitart et al.,
2017), as indicated in the corresponding figures. The latter contains forecasts out to 45 days and is available daily on a regular
latitude-longitude grid with a grid spacing of $1.5° \times 1.5°$. The data is remapped to a stereographic grid using the Climate Data
Operators (CDO; Schulzweida, 2019), hence we note that input grids other that regular longitude-latitude but supported by
CDO work equally well.



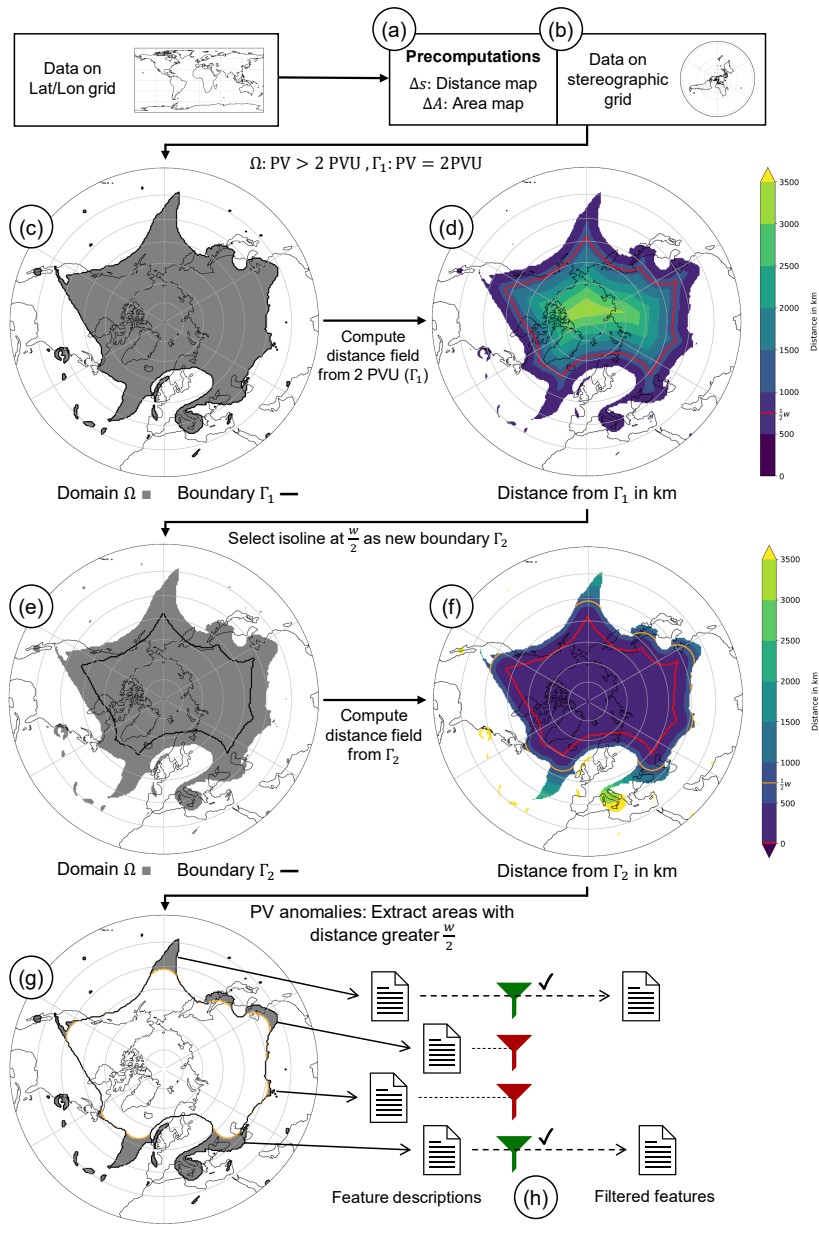

**Figure 4.** Illustration of the algorithm to identify PVSs. (a, b): Input data is mapped to a stereographic grid; precomputations are performed. (c, d): The distance field from the 2-PVU boundary is computed (cf. Sect. 3). Points at a distance equal to half the width threshold $\frac{w}{2}$ are denoted by the red isoline. (e, f): From this isoline, distances outwards are computed. Again, the isoline corresponding to $\frac{w}{2}$ is drawn with respect to the new distance field (orange). (g): Regions with a distance greater than $\frac{w}{2}$ from (f) are extracted. They form anomalies in the field. (h): The anomalies are labeled and filtered, yielding a set of identified PVSs.





---

**Algorithm 1** Pseudocode of the identification process, both for 2-D and 3-D application. 2-D and 3-D identification differ with respect to distance measure (scale analysis for 3-D) and filtering strategy (additional area based filtering for 3-D); cf. Sect. 6.

---

**Input:** $d_{PV}$: PV field, $w$: width threshold

$d_{PV}^{proj} \leftarrow \text{PROJECTION}(d_{PV})$          ▷ Project data to stereographic grid (Fig. 4b)

$\Omega \leftarrow \text{GETDOMAIN}(d_{PV}^{proj} > 2\,\text{PVU})$          ▷ Set stratospheric air mass as domain (Fig. 4c)

$\Gamma_1 \leftarrow \text{GETBOUNDARY}(d_{PV}^{proj} = 2\,\text{PVU})$          ▷ Determine dynamical tropopause (Fig. 4c)

$d_{inner} \leftarrow \text{DISTANCES}(\Gamma_1, \Omega, \Delta s)$          ▷ Compute distances from boundary into domain (Fig. 4d)

$\Gamma_2 \leftarrow \text{GETBOUNDARY}(d_{inner} = \frac{w}{2})$          ▷ Extract boundary where distance equals half the width threshold $\frac{w}{2}$ (Fig. 4d, e)

$d_{outer} \leftarrow \text{DISTANCES}(\Gamma_2, \Omega, \Delta s)$          ▷ Compute distances from $\Gamma_2$ into domain (Fig. 4f)

objects $\leftarrow \text{LABEL}(d_{outer} > \frac{w}{2})$          ▷ Extract and label areas with distance greater than $\frac{w}{2}$ (Fig. 4g)

**return** $\text{FILTERED}$(objects)          ▷ Filter anomalies by length or other heuristics (Fig. 4h)

---

## 4.2 Computation of the distance field

Before executing the identification strategy, the distance field $\Delta s$ (as introduced in Sect. 3) has to be computed (see Fig. 4a). This field is of importance for the core functionality of the identification, as well as to produce metrics for each PVS. Since the values of the distance field do not depend on the input data itself, this can be done as a precomputation step, and cached for future use.

In this study, we use a polar centered stereographic grid with a projection pixel spacing of 75 km and a total size of $340 \times 340$ pixels. This configuration covers one hemisphere completely. A more fine spacing (e.g., 50 km with a size of $510 \times 510$ pixels) increases memory consumption and run time significantly. The resolution of the projection should be chosen to match the resolution of the input data, i.e. the pixel footprints should cover similar areas. For our ERA5 analyses, the mentioned resolution is in that regard sufficient.

Furthermore, as additional metric, the *area map* $\Delta A$ is computed in this step as well. This map contains for each pixel in the projection the spherical area it projects. It can be approximated by the square of the distance map $\Delta A = (\Delta s)^2$. The area map can be used after the identification to calculate the centroid of a given PVS more accurately by taking area distortions into account.

## 4.3 Identification algorithm

Inputs for the algorithm are the field of PV data, and the width threshold $w$. It will become clear that this parameter leads to an identification of PVS up to a maximum width of $w$ km. In our study, we use a value of $w = 1500$ km, a width threshold used widely in existing techniques like the one by Wernli and Sprenger (2007). More input parameters are not required for the 2-D identification. In a first step, the stratospheric body of air is extracted from the projected data set using the previously introduced threshold of 2 PVU. This results in a binary field as seen in Fig. 4c.





The next step is called *erosion*, which is based on the respective morphological operation and encapsulates the main idea of the strategy, as introduced in Sect. 2. The domain $\Omega$ is defined as the stratospheric air mass, and the boundary $\Gamma_1$ as
the dynamical tropopause, as shown in Fig. 4c. Starting from the boundary, the distances from this boundary (dynamical tropopause) to every point in the stratospheric air mass is computed by solving Eq. 4 using the FMM as outlined in Sect. 3. Figure 4d illustrates this step and shows the color coded distance field. These are the minimum distances from the boundary $\Gamma_1$ to each grid point, respecting the boundaries of the PV field.

After that, areas that are $\geq \frac{w}{2}$ km away from the boundary are extracted, this area resembles the "inner core". Higher values
for the width threshold $w$ would remove more areas along the tropopause and thus smoothing the inner core further. Generally, this inner core may contain multiple disjunct areas. In these cases we pick the biggest one as the core to proceed with, defined by its area. The outer boundary of the identified inner PV core is defined as new boundary $\Gamma_2$ (red line in Fig. 4d) and the original PV field is kept as the domain $\Omega$.

The *dilation* operation resembles in morphology the second basic operator, as introduced in Sect. 2. The contour of the
previously defined inner core is used as input boundary, and distances outward with respect to the domain $\Omega$ are computed. Keeping this domain is necessary to measure distances following the stratospheric domain with respect to the distance map (as illustrated in Fig. 2). The distances can be seen in Fig. 4f. Areas of the dilated field with a distance $\geq \frac{w}{2}$ are extracted again (orange line in Fig. 4f). For a smooth PV field or a small value for $w$, this converges to the inverse operation to the erosion. However generally, the erosion destroys filament-like structures and cutoffs, which cannot be reconstructed that way (also
illustrated in Fig. 1d-f). The distance field will "grow" into the PVSs, providing an accurate measure for the length of a PVS.

These extracted areas resemble the identified PVS (shaded areas in Fig. 4g). Since $\frac{w}{2}$ km have been eliminated along the contour (from all sides) during the erosion step, anomalies with a maximum width of $w$ km are identified. Figure 5 shows outputs of the algorithm for different $w$. Higher values for this parameter lead to the identification of wider anomalies, as well to further extension of anomalies towards the main reservoir. The identification of cutoffs is not sensitive to parameter changes.
Finally, the identified anomalies are filtered. We separate the filtering into two steps.

1. *Area based filtering.* This step can be skipped for the 2-D identification, but will be of importance for the 3-D case as outlined later on. Areas identified by the dilation step (Fig. 4g) can be excluded, for example based on spatial information and surroundings.

2. *Object based filtering.* After the area based filtering, the identified anomalies are labeled as PVS by cohesiveness, and
are referred to as objects hereafter. For each object, a *feature vector* is computed consisting of various characteristics (see below). Then, objects can be filtered based on heuristics defined on these characteristics (Fig. 4h). Initially, our approach identifies both reasonably sized anomalies as well as very small ones (see e.g., Fig. 4g). Thus, we only keep PVSs that satisfy a certain length. We use a length threshold of 1000 km, which is based on the 2000 km threshold used by Wernli and Sprenger (2007) for the minimum length along the contour. Depending on the use case, other metrics can be used
for filtering as well.



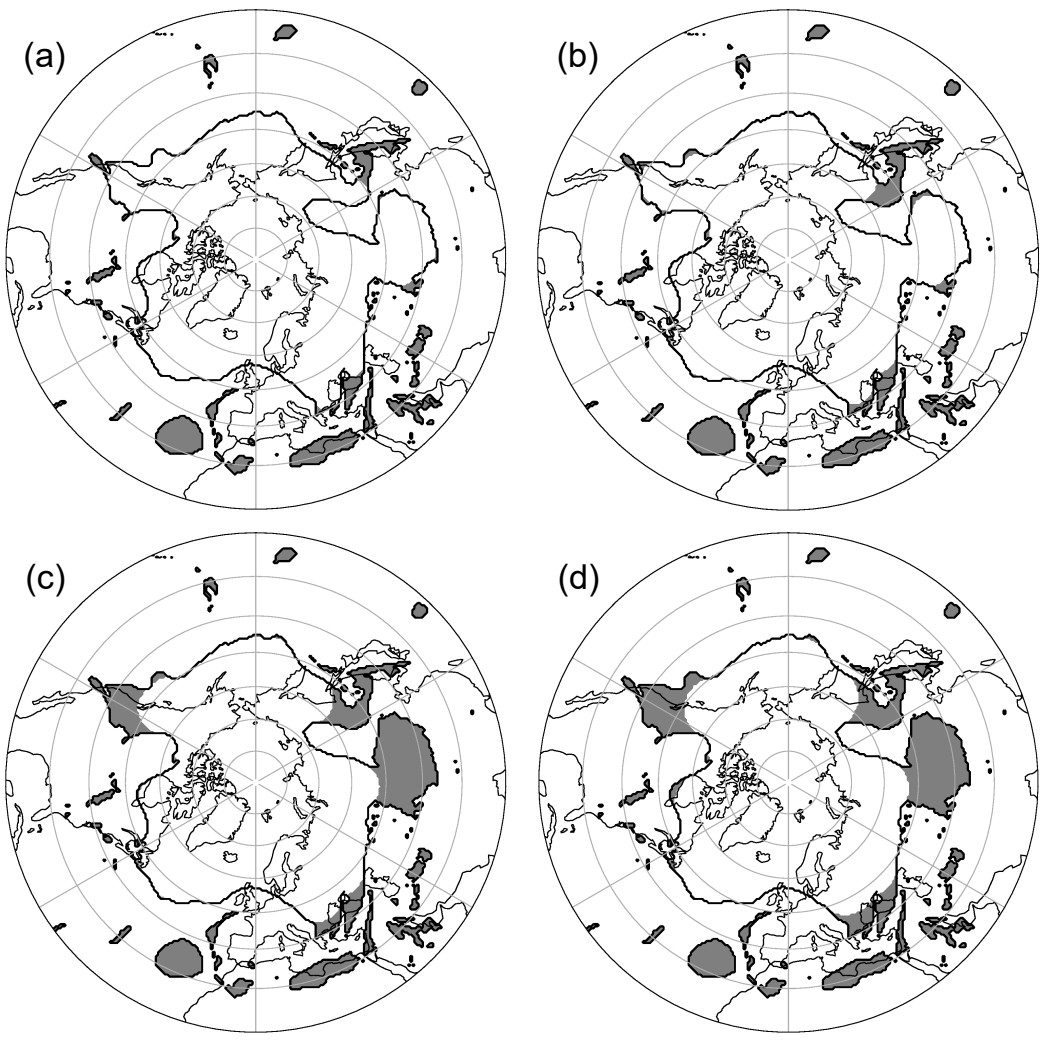

**Figure 5.** Sensitivity of the identification process to the width threshold $w$. Here, $w$ equals (a) 500 km; (b) 1000 km; (c) 1500 km; (d) 2000 km; using the example of the 330 K isentrope from the ERA5 reanalysis at 14 July 2015, 00 UTC. Results are not filtered to highlight the effect of the parameter. Our default configuration is (c).





### 4.4 Feature Vectors

Here, some elementary measures for each of the identified PVSs are introduced. This list is not exhaustive and can be extended with more metrics. For an identified object $\mathcal{O}$ containing the pixels (or area) $(x,y) = \mathbf{x} \in \mathcal{O}$, we compute:

- The object's area $A_\mathcal{O}$ as integrated pixel area using the precomputed area map $\Delta A$:

$$A_\mathcal{O} = \int_\mathcal{O} \Delta A(\mathbf{x})d\mathbf{x} \approx \sum_{(x,y)\in\mathcal{O}} \Delta A(x,y). \tag{5}$$

- The length $l_\mathcal{O}$ of a streamer, which is defined as the maximum distance $d$ emerging $\Gamma_2$ present in the anomaly $\mathcal{O}$, see Fig. 4f, minus $\frac{w}{2}$:

$$l_\mathcal{O} = \max_{\mathbf{x}\in\mathcal{O}} d(\mathbf{x}) - \frac{w}{2}. \tag{6}$$

This is a very useful metric since our strategy yields a robust length measure, while most other identification strategies only measure the length along the 2-PVU contour.

- The object's average PV, weighted by the area measure at each point:

$$PV_\mathcal{O} = A_\mathcal{O}^{-1} \int_\mathcal{O} PV(\mathbf{x})\Delta A(\mathbf{x})d\mathbf{x} \tag{7}$$

- The object's centroid as weighted average of the position over the area $\Delta A$ and the PV itself:

$$C_\mathbf{x} = (\int_\mathcal{O} \Delta A(\mathbf{x})PV(\mathbf{x})d\mathbf{x})^{-1} \int_\mathcal{O} \mathbf{x}\Delta A(\mathbf{x})PV(\mathbf{x})d\mathbf{x} \tag{8}$$

- The main axes of the object. Evaluating the eigenvalues and eigenvectors of the covariance matrix of the second order moments define the main axes of the shape (see Mukundan and Ramakrishnan, 1998). Image moments themselves are quantifiable metrics to evaluate visual information present in an image, like shape, position and orientation. Main axes form the best-fitting ellipsoid to the data, see Fig. 6. Furthermore, the major main axes can be seen as the general orientation of the object, thus the horizontal tilt of a PVS in 2-D data sets. These computations can be applied in 3-D as well, but interpretation gets increasingly difficult.

## 5 Evaluation and comparison in 2-D

Before extending the idea to 3-D data sets, the introduced algorithm will be evaluated. For this purpose, the identification strategy by Wernli and Sprenger (2007); Sprenger et al. (2013, 2017) (hereafter abbreviated by WS07) will serve as benchmark. Their strategy is well established, used in various follow-up work, and considered as state-of-the-art.

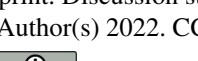



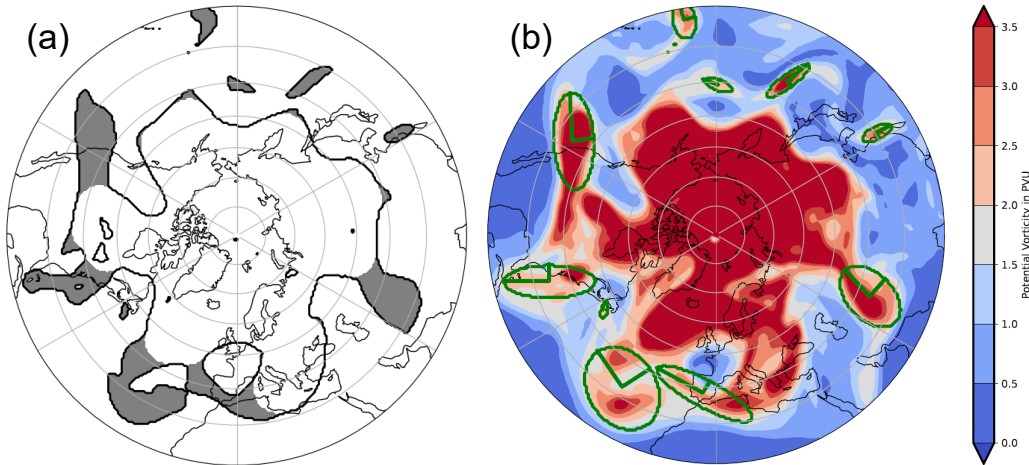

**Figure 6.** 200–500 hPa layer-averaged PV based on the S2S prediction database (Vitart et al., 2017) initialized and valid at 7 May 1998, 00 UTC. (a) PVSs identified using the default configuration without filtering the results; in (b) filtered PVSs using a length threshold of 1000 km, approximated by best-fit ellipses, with raw PV values shaded in the background. Ellipses are defined by their computed main axes.

The PVS identification strategy by WS07 is motivated by the thin and elongated nature of PVSs. They identify the outermost 2-PVU contour, and connect points along it with a spacing of about 30-50 km. This step returns a polygon resembling the outermost dynamical tropopause for the selected isentropic level. Then, the vertices of this polygon are pairwise compared regarding their direct spherical distance $d$ and length along the contour $l$. For small $d$ and big $l$ ($d < 1500$ km and $l > 2000$ km according to Sprenger et al. (2013), also with a set threshold for their aspect-ratio $l/d$), these points are regarded as streamer base points, and the enclosed shape by the polygon vertices in between the base points form a PVS.

Comparing the algorithm parameters, our width threshold $w$ has a similar interpretation to the $d$ in the work by WS07, so we set $w = d$ to evaluation purposes. This choice is also confirmed by climatological analyses we conducted, showing an inordinate increase in detected structures for smaller values of $w$. While their strategy has the additional $l$ parameter (length along contour), we do not set length restrictions in the first step, but filter the identified PVSs later based on a length threshold. This filtering parameter can be refined and changed depending on the use case.

Figure 7 shows the identification results in a 2-D PV field for both techniques. This example represents one of the predominant instances where we expect similar results. As shown in the figure, both techniques indeed identify the same structures as streamers. These also resemble the structures an educated user would identify as such. However, the shapes slightly differ: While the method by WS07 (Fig. 7b) identifies streamer base points and connects them in a straight line, our approach naturally creates roundings at these spots. These roundings reveal the boundary on what is considered as an anomaly in the field by our algorithm. The amount of detected anomalies is similar to the climatological results presented in their study, as supported by analyses we conducted.



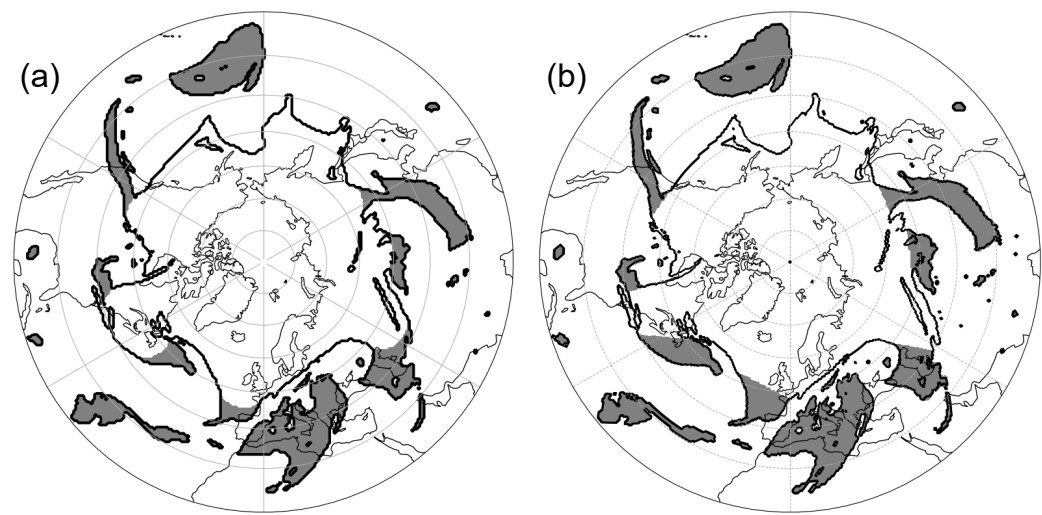

**Figure 7.** Comparison of the identified PVSs based on the 340 K isentrope from the ERA5 data set at 24 July 2015, 00 UTC. (a) Results of our strategy using the default configuration; and (b) by the algorithm from Wernli and Sprenger (2007). The anomaly over East Asia will be investigated in the case study later on.

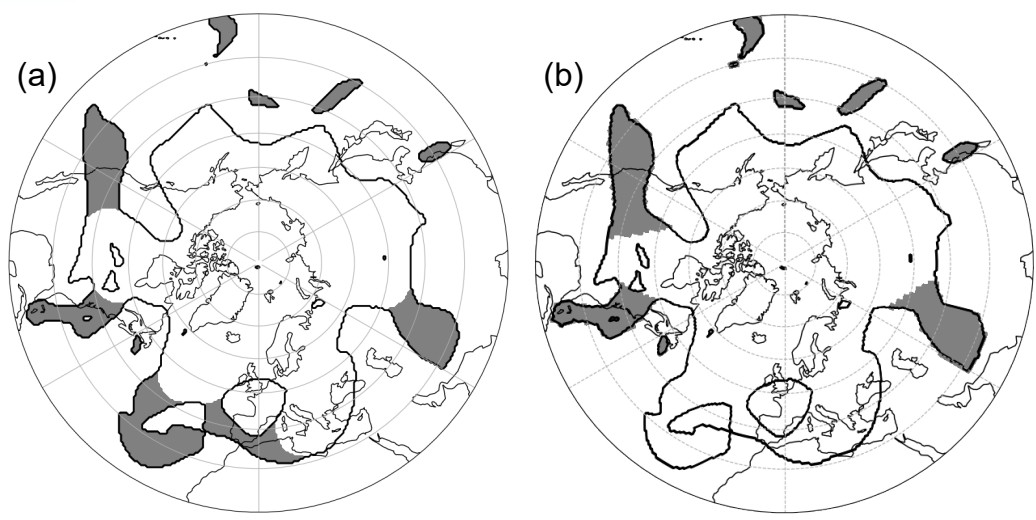

**Figure 8.** Same as Fig. 7, but with the data set used in Fig. 6.





Figure 8 on the other hand shows a more complex example demonstrating the improvements gained by considering the full PV field compared to only the outermost 2-PVU contour. Over Europe and the Eastern Atlantic, two PV intrusions (one cyclonic and one anti-cyclonic) are being observed at this specific time step. These structures are not identified by the approach from WS07, ours on the other hand is capable of identifying them. By only considering the outermost 2-PVU boundary, their algorithm only spot a broad trough from the Central Atlantic all the way to the Black Sea without taking into account the tropospheric air encapsulated in the stratospheric reservoir. With respect to our strategy, the dilation step does not reconstruct these areas, identifying them thereafter.

As introduced, we filter PVSs that do not satisfy a length threshold of 1000 km, which is comparable with the 2000 km contour length used in WS07. Note that our identification strategy does not have to be re-executed when changing the length threshold, instead the filtering step returns a different subset of identified streamers. Furthermore, our length calculation yields a precise measure for the streamer, from the reservoir base to its tip. Contour length techniques can suffer from inconsistencies introduced by noise. All in all, our strategy generally leads to additional identified streamers, especially in complex environments.

For a set test case, the computation of anomalies takes a similar amount of time. Regarding our strategy, this time also includes the computation of feature vectors (e.g., best fit geometry and centroids) for all detected structures. Nevertheless, due to different approaches in preprocessing, different programming languages and different use of parallelism, a quantifiable comparison of the run times proves to be difficult. For our algorithm, 85% of the computational time is required to project the data onto a stereographic grid. Analyzing bigger data sets, data level parallelism (e.g., Single Instruction Multiple Data (SIMD) processing) can highly improve the efficiency of projecting data from multiple isentropic levels and multiple time steps. Projecting a 3-D data set with 50 isentropic levels takes only twice the time compared to a single isentropic level, signifying a very high speedup.

Similar to WS07, we currently do not consider the origin of a PVA as a filtering strategy. For example high PV of non-stratospheric origin (e.g., in tropical cyclones (Molinari et al., 1998) or related to deep moist convection (Weijenborg et al., 2017)) could also be identified as a PVS or cutoff. More sophisticated time series and algorithms are needed to distinguish the origin of a PV filament, e.g., using specific humidity thresholds as suggested by Škerlak et al. (2015). Our concept of feature descriptions can give some insight and provide quantified values for each identified anomaly, but this problem goes beyond the scope of this work.

# 6 Extending the principle to three dimensions

## 6.1 Data

For the 3-D identification, ERA-5 data on model levels (here we use levels 40-137) are interpolated to isentropes with a vertical spacing of 2 K. We consider the region between 290 K and 380 K. The vertical spacing of 2 K proved to be a reasonable trade-off between computation time and disk usage on one hand, and representing the vertical structure of the PV field on the other hand. Our experiments showed that coarser spacings (e.g., 5 K) hide dynamically relevant and fine features in the





vertical structure, while a finer spacing would increase memory and computational cost significantly without adding much more relevant structures. Using pressure or model levels would be an alternative worth consideration, since it is natively available in higher vertical resolution as model output. Therefore, it would not require the interpolation on isentropic levels, but would reduce the interpretability of the PV structures, especially when evaluating them using cross-sections or development

over time.

## 6.2 Strategy

There are multiple ways to extend the in Sect. 4 introduced algorithm to 3-D. A straightforward idea consists of applying the 2-D identification to each isentropic layer individually and stacking the results on top of each other to create 3-D objects (e.g., Portmann et al. (2021) for cutoffs). Applying our algorithm in a similar way to 3-D data sets would be trivial, but would not

support the idea of including the information of the 3-D shape around each given point. For example, the use of thresholds (e.g., minimum contour length or aspect-ratio) in 2-D identification can lead to artifacts when applied to fine resolved vertical levels: A PVS might be just above a given threshold and therefore being identified on specific isentropic levels, but not on a level in between where the structure is just below this set threshold. This generates so-called popping artifacts.

Our algorithm is designed in a manner that it can be executed on 2-D and 3-D data in a similar way. Note that the basic ideas

for erosion and dilation (see Sect. 2) can be applied independently on the number of dimensions and can therefore be used the same way. Existing identification techniques (see an overview of them in Papin et al., 2020) rely on following the 2-PVU boundary, and searching points along this boundary meeting certain criteria. However in 3-D, there is no unique direction to follow the dynamical tropopause, making such techniques not applicable. In our study on the other hand, the 2-D boundary $\Gamma$ introduced in Sect. 4 can be extended to a 3-D boundary (isosurface) and handled the same way.

The distance measure introduced in the 2-D identification needs to be expanded to another dimension to be applicable to a 3-D field. An exact distance approach would compute the height of every grid point in the atmosphere given its current state, or based on a standard atmosphere model. Due to too high computational effort we keep isentropic levels as vertical distance measure and combine the mismatching units in the horizontal (euclidean distance) and vertical (potential temperature) direction motivated by a scale analysis. In a true-scale visualization, 3-D PV anomalies would be very flat. Bithell et al. (1999) used a

vertical scaling factor of about 100 in their z-axis to achieve clearly visible 3-D structures in the PV field. We approximate the horizontal scale of PVAs in the magnitude of 1000 km and their vertical scale in the magnitude of 10 K. As a result, we stretch the vertical dimension by factor $100\,\mathrm{km\,K^{-1}}$ to achieve a similar scaling like Bithell et al. (1999), and use an euclidean measure when computing these stretched distances. This helps our algorithm to "work" along all dimensions simultaneously rather than focusing on the horizontal dominant level. For the algorithm, we use the same width threshold $w$ as before.

Figure 9 shows an example of 3-D visualizations of the 2-PVU contour and the identified anomalies. For visualization purposes, we use the stereographic projection to emphasize the circulating nature of the flow. The isosurface represents an isovalue of 2 PVU (tropopause) and is shaded by isentropic height. This is analogous to the boundary $\Gamma$ in the 2-D case. As expected, the tropopause has a lower height towards the North Pole (beige shaded area), and a sharp gradient is situated in the mid-latitudes where the jet stream is located (Koch et al., 2006). Visible holes in the isosurface resemble tropopause levels





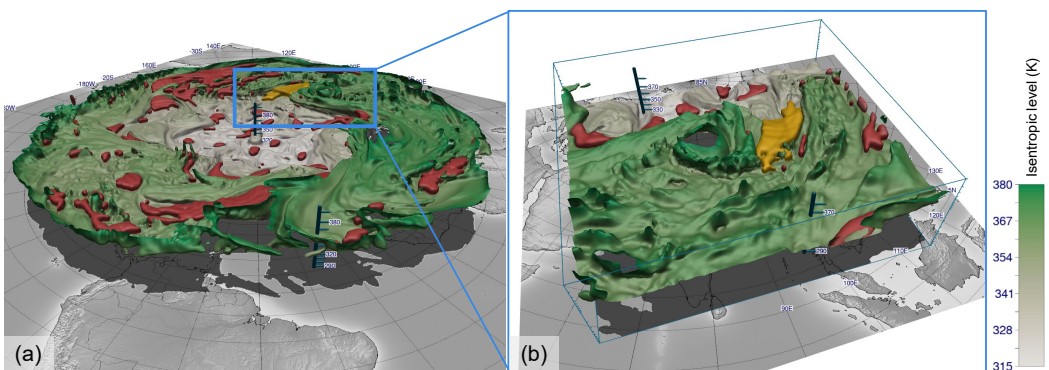

**Figure 9.** 3-D visualizations of the dynamical tropopause (defined by the 2-PVU threshold), which is shaded by isentropic height. Red and yellow are the identified anomalies. The yellow one is emphasized for the case study later on. (a) The entire northern hemisphere is visualized; in (b) focused view of southeastern Asia, which is investigated in the case study in Sect. 7.

beyond our domain of 290 K to 380 K. The red and yellow regions (latter emphasized for the case study) are being identified as PVAs. Especially for our case study later on, we use the zoomed-in area around East Asia (Fig. 9b) to analyze the event.

Generally, the same filtering process as in 2-D is applied. The goal is primarily to keep as many detected features as possible. The user can then decide which of these features are of importance for a specific use case and filter these by their feature descriptions accordingly. The centroid can be computed in 3-D, while we replace the area measure with a volume measure (in

km$^2$K, or approximated in km$^3$ using a standard atmosphere model). Some of the feature descriptions introduced in Sect. 4.4 are not computed in 3-D due to misleading interpretations introduced by the scale analysis. For example, the length measure implicitly contains the applied scale factor of $100 \, \text{km} \, \text{K}^{-1}$. As stated above, this scaling is required for the 3-D functionality. For analyses on these features, we suggest to use measures that are easy to interpret, like the object's bounding box or volume. Typical use cases for filtering involve setting thresholds for position, size, or values of other fields in the same area (e.g.,

humidity).

However, in 3-D, we have to introduce an area based filtering strategy (cf. Sect. 4.3). Fig. 10a shows the raw and unfiltered results of our identification applying the strategy described in this section. Thus, despite their distinct nature on 2-D isentropic levels, these 2-D PVS are connected to each other via 3-D structures. Actually, this behavior is expected from a globally coherent flow structure. However, it is desired to have clear and distinct anomalies, so they can be described individually for

statistical analyses. Therefore, we filter the identified anomaly areas to get a set of detached and separated objects. We discard identified areas with a vertical extension of smaller than $b = 6 \, \text{K}$. This specific value has been chosen as a good compromise of grasping clear features, while separating them at areas of weak vertical extent. Illustrated in Fig. 10b, c are identification results using different values of the extent threshold $b$. Figure 10b shows our default configuration.





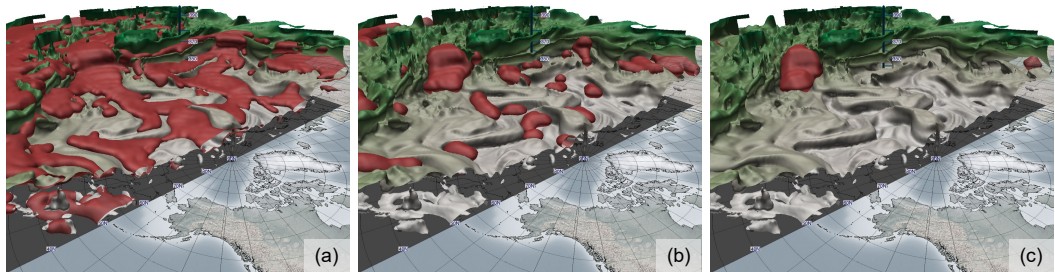

**Figure 10.** Effect of filtering anomaly candidates with respect to the extent threshold $b$. Areas of anomalies with a vertical extent smaller than $b$ are omitted from the identification, this leads to clear and separated features. (a) $b = 0$ K (no filtering); (b) $b = 6$ K; (c) $b = 12$ K. As specified in Sect. 6, our default configuration is 6 K.

### 6.3 Implementation Details

Our feature identification strategy is implemented in a Python framework for general identification and tracking of meteorological structures. This framework is part of the *enstools* Python package (Redl et al., 2018), which provides routines for meteorological data processing. Identified features can be saved to disk in an efficient binary format or as humanly readable JSON format. The algorithm itself relies on the CDO Python wrapper (Schulzweida, 2019) for projections. The *scikit-fmm* toolkit (Furtney, 2019) provides an implementation for the FMM algorithm, which has been adapted to fit our requirements. Our software is open-source and can be accessed as outlined in the code availability section.

On a test system (AMD Ryzen 5 2600X), the identification process for a 3-D data set with 46 isentropes (290 K to 380 K) and a horizontal grid spacing of $0.5° \times 0.5°$ for the northern hemisphere takes 5.9 s, where projecting the data takes about 23% of the total time. The computation time heavily depends on what features are being computed for the individual objects (e.g., main axes, volume). Compared to the 2-D identification, where one isentropic level took about 1 s to process, this 3-D identification reveals a big speedup regarding the amount of data being processed.

### 7   Case Study: Precursors of an extreme precipitation event affecting northeastern Vietnam

We evaluate our identification method along a case study. The selected extreme precipitation event affected northeastern Vietnam in late July and early August 2015. This event has been analyzed in detail by Van der Linden et al. (2017), particularly focusing on the predictability of the event in ensemble forecasts. According to Van der Linden et al. (2017), predictability of the event was strongly related to the presence and location of an upper-level trough in ensemble forecasts. The trough was diagnosed using 200 hPa geopotential height and upper-tropospheric PV fields. Here, we want to take a closer look into the precursors of this trough, and to analyze whether our novel technique can help to improve the analysis of the dynamics leading to the event. Different configurations of the algorithm are considered and our intent is to shed light on the 3-D structures. Visualizations are made using the open-source visualization framework Met.3D (Rautenhaus et al., 2015b). The visualized



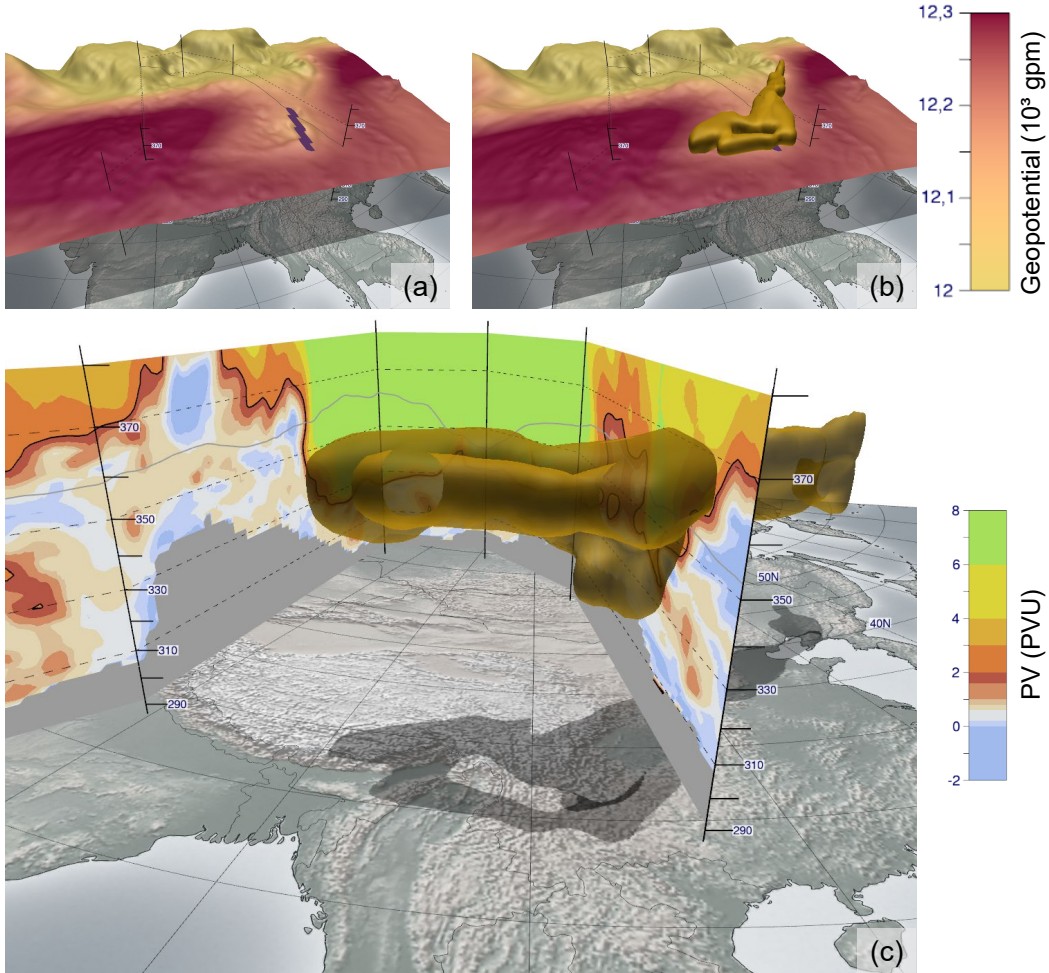

**Figure 11.** Comparison of the location of the anomalies defined by the 200-hPa geopotential height and by the dynamical tropopause, at 25 July 2015, 12 UTC. (a) Shown is the 200 hPa isosurface on isentropic levels, shaded by geopotential height. The purple line is the trough identified by Van der Linden et al. (2017) in a 2-D environment, cf. their Fig. 7. (b) Same as (a), but the identified PVA has been added. Its location clearly coincides with the geopotential trough and the identified trough axis. (c) A vertical cross-section of the area along the section indicated in (a, b), and the identified PVA. Shaded on the cross-section is the PV, where the black contour highlights the 2-PVU boundary. The gray line is the 200-hPa contour, thus a cut of panel (a).

data is based on the ERA5 reanalysis as introduced in Sect. 6.1, using a 6-hourly temporal resolution. To assess the rainfall associated with the event, the 6-hourly rainfall totals centered around the respective time steps and based on the satellite-based NASA Global Precipitation Measurement (GPM) Integrated Multisatellite Retrievals for GPM (Huffman et al., 2015) product were computed. This data is available at a horizontal grid spacing of $0.1° \times 0.1°$.





Between 25 July and 3 August 2015, record-breaking precipitation amounts of more than 1000 mm were observed at the
coast of northeastern Vietnam. According to the analysis by Van der Linden et al. (2017), intense and long-lasting rainfall was
mainly related to a surface low over the Gulf of Tonkin, which slowly moved westwards. Van der Linden et al. (2017) provided
evidence that the formation and westward movement of the surface low was related to a subtropical trough that was located
over southern China slightly to the northwest of Vietnam. Figure 11a, b shows the 200 hPa isosurface in an isentropic field.
The isosurface is shaded by geopotential height. It also shows the location of the identified trough axis by Van der Linden et al.
(2017), using a 2-D analysis. In Fig. 11b, the identified PVA has been added. Clearly, the geopotential anomaly coincides with
the position of the trough axis and the PVA. Fig. 11c shows for the same point in time a corresponding vertical cross-section,
which will be used for the PV analysis in this case study. The location of the PVA matches the PV intrusion over southeastern
Asia.

## 7.1 PV analysis

To examine the precursors of the extreme event, we visualize the 3-D PV environment over eastern Asia for five different time
steps in Fig. 12. These panels provide snap-shots, starting four days prior to the extensive rainfall. For an additional depiction
of the event based on an extended time frame, also including intermediate time steps, we provide an animation of the synoptic
evolution of the PVA over time in a supplementary video at Fischer et al. (2021b).

At 21 July 2015, a wide trough is located over eastern China, which is below the thresholds for our identification strategy in
2-D and 3-D to be detected as a PVA. Over the following days, a high tropopause area is moving from the Arabian Peninsula
towards the Tibetan Plateau, further illustrated in the supplementary animation (Fischer et al., 2021b). In the visualization,
this anomaly is characterized by a "hole" in the 2-PVU isosurface, and a high tropopause on the western side of the vertical
cross-sections in Fig. 12a-c. Holes in the tropopause are heights beyond our domain that has an upper limit of 380 K. From 22
July 2015, 00 UTC onward, our algorithm identifies a PVA related to the event over Vietnam (shaded in yellow in Fig. 12).
The high tropopause over the Tibetan Plateau shows to be stationary over multiple days, leading to intense advection of high
PV air southwards along its eastern flank. This leads to an intensification of the identified anomaly until 24 July (Fig. 12a-c).
After this, the influence of the Tibetan high becomes weaker, cutting the intrusion of high PV air from the north. The identified
structure partially cuts off horizontally from the main reservoir (Fig. 12c). During this process, the centroid of the PVA moves
southward from around 30°N over mainland China to around 20°N just west of Vietnam. This is clearly visible by the PVA
moving southwards along the cross-section from Fig. 12a-c. The period afterwards coincides with the most intense rainfall over
the coast of northeastern Vietnam as indicated by the 6-hourly precipitation rate (cp. Fig. 5 in Van der Linden et al. (2017), and
emphasized blue boxes in Fig. 12b-d). Over the following days some dynamical coherence to the main tropopause fold can
still be observed (Fig. 12d, e). Finally, on the first days of August the PVA dissipates (see Fischer et al., 2021b).

Comparison with the results of Van der Linden et al. (2017), who used 200-hPa geopotential height and 500–200-hPa
vertically averaged PV to detect the subtropical trough (their Fig. 7), clearly illustrates the advantage of our novel 3-D approach.
Using vertically averaged PV, and single-level geopotential height, the trough could not be identified after 25 July, and 29 July,





**Figure 12.** The synoptic-dynamic development associated with the event. Here, five time steps are visualized: (a) 22 July, 12 UTC; (b) 24 July, 00 UTC; (c) 27 July, 00 UTC; (d) 30 July, 12 UTC; (e) 31 July, 12 UTC. Respectively on the left, a 3-D visualization of the dynamical tropopause shaded by isentropic height, and the identified anomaly of interest is shown. On the right, the cross-section as displayed in Fig. 11c is shown. The blue box in (b-d) emphasizes the torrential rainfall triggering the flood. An animation with an extended time frame is provided at Fischer et al. (2021b).



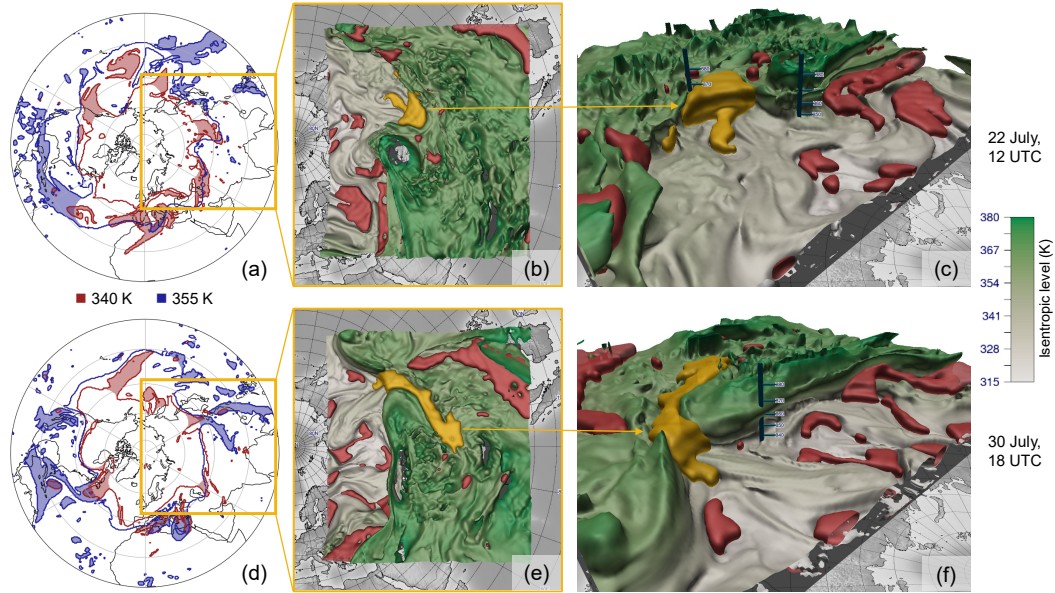

**Figure 13.** Comparison of the 340 K and 355 K isentrope, and the corresponding 3-D analysis for two time steps throughout the case study. (a): The 2-D analysis at 22 July, 12 UTC, where the anomaly is identified on 340 K (red), but not on 355 K (blue); (b): top-down view of the 3-D analysis; and (c): side view of the 3-D anomaly. (d-f): Same as (a-c), but 8 days later. The anomaly is now identifiable on 355 K, but not anymore on 340 K. Therefore, a 2-D analysis of the event considering a single isentrope is insufficient.

respectively, by Van der Linden et al. (2017). Since high precipitation amounts were also observed after 29 July and in more western parts of the country (not shown), a continuing influence of the PVA seems reasonable.

### 7.2 Algorithm evaluation

As outlined in Sect. 6, two parameters in this algorithm for 3-D cases are used, the width threshold $w$ (see Sect. 4) and the extent threshold $b$ to separate PV anomalies into clear disjunct features. For this case study, we use the default configuration of $w = 1500\,\mathrm{km}$ and $b = 6\,\mathrm{K}$.

To give an understanding of identifying 3-D structures over their 2-D counterpart, Fig. 13 illustrates two different time steps of the case study. In Fig. 13a-c, a time step early in the development of the event is shown. The yellow anomaly can be

identified as an equatorward intrusion of stratospheric air in the 3-D view. Considering identification on 2-D levels (340 K and 355 K isentropes in Fig. 13a), the structure is not identified on the 355 K isentrope due to its broad shape, but the anomaly is clearly distinct on the 340 K level. Because our 3-D technique (Fig. 13b, c) takes the whole 3-D neighborhood into account, the identified object covers both selected levels. Just looking at single 2-D isentropes individually is not sufficient to grasp the vertical structure of the PV object. Later in the event (Fig. 13d-f) the same anomaly that is clearly identifiable in 3-D can

be detected on the 355 K level, but not anymore on the 340 K level using the 2-D identification. Therefore, the PVA moved vertically upwards throughout the event, making it impossible to automatically analyze and track when using a single isentropic





level. This also falls into place when analyzing the height of the PVA's centroid. During genesis, the centroid (position weighted by volume and PV, see Sect. 4.4) of the PVA has a height of 345 K, and in later stages of the event, it is situated at over 360 K.

The case study shows as well that distinguishing between the concepts of a PVS and PV cutoff in 2-D is not applicable to the 3-D view. The vast majority of cutoffs are attached vertically to the main stratospheric reservoir, while PVS are connected typically both horizontally and vertically. Furthermore, a clear structure could often be interpreted as a streamer or a cutoff depending on the isentropic cross-section one is investigating. Visually, it can become unclear at all, which type a 3-D structure more belongs to. This supports our suggested nomenclature to take a more abstract approach and name these intrusions anomalies (PVAs).

## 8 Conclusions

In this study, a novel automated identification strategy for PVAs has been presented. It is based on sequential distance measures in the PV field to extract anomaly-like structures. The method is capable of identifying these structures in both 2-D and 3-D data sets robustly. It is based on image processing operations, which are able to detect disturbances in a multidimensional field, and has been adapted to suit the needs and requirements of the present application. Specifically, the adaptation keeps the interpretability of the process by using physical units, and compensates the distortions introduced by the particular flattened projections used. Here, we use the Eikonal equation and reinterpret it in a novel fashion to efficiently compute accurate distances in the distorted field, taking advantage of the properties of the projection. The strategy has proven well scalable for processing big data sets. The presented algorithm takes a width parameter $w$ as input, which controls the maximum width of anomalies being identified. Identified anomalies are assigned feature vectors containing metrics for the specific object, e.g., centroid, bounding box, intensity or best-fit ellipsoid. The objects are then filtered based on these metrics contained in the feature vectors.

A drawback of existing 2-D identification techniques is that their decisions are solely based on the outermost 2-PVU contour for a given isentropic level. It is shown that this hides important information about the structure of the field in some cases. By applying our algorithm, which considers the whole field, we therefore identify additional features not being easily captured by most algorithms described in the literature. The algorithm can be executed in the same fashion for 2-D and 3-D data sets since the individual image processing operations are independent of the number of dimensions. One main issue of the 3-D identification consists of clearly identifying separate anomalies in the 3-D PV field. We therefore added a heuristic that separates anomalies by detaching them at points of weakest vertical extent for a clear separation. This is important for meaningful feature descriptions afterwards.

In 3-D analysis, the algorithm provides an independent configuration for all seasons and use cases, and is therefore suitable for climatologies. In 2-D analysis on the other hand, care has to be taken of choosing a suitable isentrope for analysis during a season or for a specific event, or depending on which regimes one is interested in. Also more interesting features can be detected by considering the full 3-D neighborhood instead of focusing on a single 2-D level. The 3-D structures are more cohesive and





consistent in space and time compared to their 2-D counterparts. Popping artifacts, which appear in 2-D identification when
stacking results of individual levels, are prevented.

A case study has been investigated to show up advantages of the 3-D analysis. A trough detected using single-level geopotential height or vertically averaged PV as in Van der Linden et al. (2017), which was instrumental in the evolution of the presented extreme event, would dissipate too early when compared with the results using the novel approach. Comparisons of 2-D and 3-D identification results reveal that the 3-D object is present both over a broader vertical range and longer time period.
Especially the longer period coincides more closely with the observed rainfall than the 2D analysis by Van der Linden et al. (2017). Therefore, the novel method provides a more detailed view of the dynamical forcings of the extreme event by taking into account the vertical evolution and movement. This supports conclusions by Bithell et al. (1999). They emphasized that without a full 3-D view of the developing system the extent of features, such as tropopause folds, and their depth of penetration into the troposphere, are difficult to follow.

We put effort in describing the identified object in an abstract geometric fashion, similar to ellipses in 2-D. There, most structures can be well approximated by one (or in degenerative cases a few) simple geometric shapes. These are also clearly interpretable, e.g., the main axis yields the tilt of the streamer. For 3-D objects, we considered using subsets of quadrics (second order surfaces, see Krivoshapko and Ivanov, 2015) but recognized that the extension of 2-D fitting to 3-D data is a highly nontrivial task. To explore the geometrical structure, but also to exploratorily analyze the presented algorithm and the data set
itself, interactive 3-D visualizations proved to be an essential tool for comprehensive depictions. Derived displays, such as cross-sections, are vital to highlight dynamical processes in complex environments and to perform coherent conclusions.

In the future, this technique can be applied both to more individual use cases and to big data sets. Analyses and climatologies help to exploratorily examine the properties and behavior of PVAs in certain 3-D environments. Computed feature descriptions could be a base to find correlations and clusters in data, classifying anomalies into distinct categories. Since the algorithm is
well scalable, it can be applied to ensemble forecasts as well. However, the performance bottleneck for big data analysis is the computation of high resolution isentropic data out of model levels. Therefore, processing large data sets might require a trade-off by using a different vertical dimension even though this leads to a more challenging interpretation, especially when evaluating horizontal cross-sections.

Further research is required on certain aspects of this identification, which might lead to improved results. For individual
use cases, the tracking of these anomalies could be done by hand, whereas for climatologies automated tracking techniques are required. Ideas for further work include simple overlap tracking (e.g., Limbach et al., 2012), or a Lagrangian view (e.g., Portmann et al., 2021). The feature descriptions, which have been computed for each PVA, might serve as base for similarity measures and be used for tracking purposes.

The automated 3-D PV identification, tracking, and its application to many meteorological cases opens an avenue to study,
which characteristics of the PV objects are related to the genesis and improved forecast using statistical and machine learning approaches. For example, Maier-Gerber et al. (2021) successfully used ensemble mean and standard deviation of vertically averaged upper-level PV forecasts in the Gulf of Mexico as a predictor for subseasonal statistical-dynamical forecasting of tropical cyclone occurrence. Here, a 3-D PV object related approach holds the promise of further improvements. These ab-



stracted objects can also facilitate combinations with other displays, e.g., other identified atmospheric features, to explore
further connections.

*Code and data availability.*   The implementation of our framework is available at https://gitlab.physik.uni-muenchen.de/Christoph.Fischer/enstools-feature (where it will be updated in future studies) and archived at https://doi.org/10.5281/zenodo.5638561 (Fischer et al., 2021a). It is realized as part of the open source framework *enstools*, available at https://github.com/wavestoweather/enstools. The data sets used in this study (ERA5 reanalysis, S2S) are publicly available at the relevant cited sources.

*Video supplement.*   A video supplement showing 3-D visualizations of the synoptic evolution of our case study can be freely accessed at https://doi.org/10.5281/zenodo.5639001 (Fischer et al., 2021b). The visualizations are generated using Met.3D (Rautenhaus et al., 2015b).

*Author contributions.*   ES, AF and M Riemer proposed, supervised and administrated this study. CF designed the algorithm, implemented the software, performed algorithm analyses, did visualizations and wrote the publication. RL performed meteorological analyses on the case study and provided according results. M Rautenhaus contributed helpful input and comments regarding Met.3D and visualizations in general. 555 All authors provided feedback and critical review on the paper.

*Competing interests.*   The authors declare that they have no conflict of interest.

*Acknowledgements.*   The research leading to these results has been accomplished within the project C3 "Predictability of tropical and hybrid cyclones over the North Atlantic Ocean" of the Transregional Collaborative Research Center SFB/TRR 165 "Waves to Weather" funded by the German Science Foundation (DFG). This work uses S2S data. S2S is a joint initiative of the World Weather Research Programme 560 (WWRP) and the World Climate Research Programme (WCRP). The original S2S database is hosted at ECMWF as an extension of the TIGGE database. The GPM (IMERG) data were provided by the NASA/Goddard Space Flight Center, and archived at the NASA GES DISC. Thanks to the Institute for Atmospheric and Climate Science at ETH Zurich, in particular to Michael Sprenger and Heini Wernli, for sharing the source code of their PV identification technique (Wernli and Sprenger, 2007).



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
