# Peer review of "A novel method for objective identification of 3-D potential vorticity anomalies"

_Geoscientific Model Development, 2021_

## Referee Comment (RC1)

Review of "A novel method for objective identification of 3-D potential vorticity anomalies," by Christoph Fisher et al.

**General Comments**

This paper provides a novel technique for identifying the 3-D structure and temporal evolution of potential vorticity anomalies. Using a morphological image processing technique on a stereographic projection they identified PV candidates and filtered them for further analysis. Extension of the technique to 3-D is presented using a case study of a large precipitation event in Vietnam in 2015. The paper is generally well written and suitable for publication in Geoscientific Model Development. Below are some suggestions for improvement. The main issue is that more clarity is required in explaining the 3-D extension of the algorithm and description of the 3-D analyses.

**Specific Comments**

Line 23: May want to list some of these nonconservative processes. Also, what do you mean by "large-scale"?

Lines 163-165: How do you merge the data at the equator? Is it possible to get different results from the two mappings? Or do you just ignore this region for this study and leave the boundary problem for future work?

Line 173: Is this angle measured with respect to the center point point $(x_c, y_c)$?

Line 177: Should the exponent be 2 rather than -2. Otherwise, what does cos$^{-2}$( ) represent?

Line 184: I don't understand how $\Delta s(x_c, y_c)$ is calculated. Can you give an example or show this distance on Figure 3?

Line 196: Does $z$ represent an infinitesimal line segment between any two arbitrary points?

Line 201: What is the gradient used here? Is it a spatial gradient along the projection plane?

Figure 4 caption: I know this is an illustrative figure, but it would be helpful to know what dataset, time, vertical level, as in Figure 5, in case a reader may want to try to reproduce these results.

Lines 240-1: The threshold PV value (2 PVU) could possibly be classified as an input parameter. You don't show any sensitivity to the choice of this parameter, but clearly it could have a significant impact on your analysis. You may want to comment on this.

Line 255: What about features that are disconnected from the main part of the region, like the small filaments in the Atlantic Ocean. How do you calculate their distances since no path from the inner boundary is available that stays within the domain?

Line 254: Is the contour here defined as the boundary of the object? If the object is connected to the main reservoir do you include the line segments where the object and the reservoir intersect?

Section 4.4: This is a nice description of various metrics, but it seems that they are not really used much in the paper. It would be nice to have some examples of these calculations for different features in one of the 2-D examples.

Figure 6: It is interesting that this figure is based on pressure layer-averaging rather than isentropic level. Is there a reason for this? Also, it is unclear how the individual objects are distinguished here. For example, the feature over the Atlantic Ocean appears to have two distinct PV anomalies. Are they both part of one object?

Lines 327-328: Regarding the length calculation, is this only for features that are connected to the main reservoir? What about the cutoff features in the ocean? How do you calculate the length of those features for the filtering algorithm to use, since they are disconnected? This is related to an earlier question.

Lines 320-323: I don't understand this sentence. How does their algorithm identify a broad trough? Are your referring to the 2 PVU contour on Figure 8b? And what is the tropospheric air encapsulated in the stratospheric reservoir? Can you point to that in the figure?

Line 367: The first sentence seems to be missing some words "extend the in Section 4"

Line 370-379: The description of the 3-D extension of the algorithm seems to be lacking important information for the reader. Here are a few questions and comments.
- In 2-D you have boundary defined by the 2 PVU contour and domain defined by the area enclosed by that contour, which includes PV values greater than 2 PVU. In 3-D the boundary is now the 2 PVU surface, but what is the domain? Is it the 3-D region where the PV value is greater than 2 PVU? So generally you are calculating distance upwards from the 2 PVU surface?
- In 2-D you calculate the distances constrained by lines that remain in the domain. But in 3-D are the distances still required to stay in the domain? How is that done?
- This 3-D calculation takes into account both vertical and horizontal displacements, and in the vertical you use a scaling between potential temperature and height. Couldn't you use the geopotential height as the approximate vertical position of each grid point?
- Are there complications in your algorithm when there are multiple tropopauses in one vertical column? How does the algorithm handle these?
- It could be that one additional figure showing how the process works in 3-D with a simpler shape than in Figures 9 and 10 would help the reader to understand this extension. Could you make plots similar to Figure 4, but with the 3-D domain? For 3-D, it is not obvious what is the domain, the two boundaries, the direction for which the distances are calculated, etc.

Figure 9: It looks like the anomalies are above the dynamical tropopause, extending into the higher PV stratosphere. Is this an optical illusion? Also, I wondered why the long streamer in the central lower part of Figure 9a is not identified as an anomaly. Is this too thin in the vertical to meet the 6 K threshold?

Line 382: Would this be analogous to $\Gamma_1$ in Figure 4a? What would $\Gamma_2$ look like?

Line 389-390: Centroid and area don't seem to be calculated anywhere. Can you give an example?

Line 400: This says you filter the areas to get detached and separated objects. It seems arbitrary that the 6K cutoff would actually separate the regions into dynamically distinct features. Won't this artificially cut off certain features that should be connected?

Line 469-470: The yellow anomaly is an equatorward intrusion of stratospheric air. Since tropopause height increases equatorward, wouldn't the anomaly lie under the tropopause, like a tropopause fold?

Figure 13 gives the appearance that the anomaly is above the tropopause. Maybe it would help to explain exactly what the isosurfaces of the anomalies represent in your 3-D diagrams.

**Technical Corrections**

Figure 2 caption: Do you mean "FMM" here, not "FFM". Also, do you want to capitalize "marching method"?

Line 246: replace "is" with "are", since the noun is plural (distances)

Line 281: Do you mean distance d emerging "from" ...

Line 305: Should "form" be "forms", since the noun is singular (shape)

Line 323: "spot" should be "spots"

---

## Author Comment (AC1)

**RESPONSE TO REVIEWERS**
**A novel method for objective identification of 3-D potential vorticity anomalies**
**10.5194/gmd-2021-424**

We would like to thank both reviewers and the editor for their time spent on reviewing our manuscript and their thoughtful comments that helped to improve the article. Below, we provide detailed point-by-point replies (blue font color) to each comment. Technically, citations of text passages are in italics. Text changes are highlighted in yellow. Line numbers refer to the submitted preprint.

**Reviewer # 1:**

General comments
The main issue is that more clarity is required in explaining the 3-D extension of the algorithm and description of the 3-D analyses.

We thank both reviewers for their valuable input regarding the 3-D extension of the algorithm, especially the lack in clarity regarding the methodology. The 3-D extension is a main novelty in this study thus we have thoroughly revised the according text sections. A new figure is introduced to make the step from 2-D to 3-D more intuitive for the reader. In this general comment, we address the text changes, the new figure, and the specific comments by both reviewers regarding the 3-D methodology.

We kept the first part of Section 6.2 conceptually unchanged but rephrased some sentences for clarification purposes. This part provides a brief introduction and outlines why simply stacking results of the 2-D identification would not yield satisfying results. The new Section 6.2 starts as followed:

*"There are multiple ways to extend the algorithm introduced in Sect. 4 to 3-D. A straightforward idea consists of applying the 2-D identification to each isentropic layer individually and stacking the results on top of each other to create 3-D objects (e.g., Portmann et al. (2020) for cutoffs). However, applying our algorithm in such fashion would not include all information available in the full 3-D environment at a given point. For example, the use of thresholds (e.g., minimum contour length or aspect-ratio) in a 2-D identification strategy can lead to artifacts when applied to multiple vertical levels: A PVS might be above a given identification threshold on two specific isentropic levels, but below this threshold on a level in between these two, therefore not identifying this structure on this intermediate level. This leads to unwanted vertical gaps in the detected structure."*

Then, after Line 363, to improve perception and clarity, we first introduce the 3-D visualization before giving a more technical description of the strategy to extend the algorithm to 3-D. We refer to the unchanged Figure 9 in this passage to achieve this. We also add visual descriptions of anomalies to give the reader a general idea of the shape and structure of them. Subsequently, the technical part will be amended with a new figure, therefore introducing this form of display beforehand is vital.

"*Figure 9a shows a 3-D visualization of the tropopause (defined by the 2 PVU isosurface) and the identified anomalies. Like for the previous 2-D visualizations, we use a stereographic map projection to emphasize the circulating nature of the flow. The tropopause is shaded by isentropic height. The air mass below (above) this isosurface is classified as tropospheric (stratospheric) air. As expected, the dynamical tropopause has a lower height towards the North Pole (beige shaded area), and a sharp gradient is situated in the mid-latitudes where the jet stream is located (Koch et al., 2006). Visible holes in the isosurface resemble tropopause levels beyond our visualization domain of 290 K to 380 K. The red and yellow regions (latter emphasized for the case study) are the identified PVAs. As visible in Fig. 9 and supported by the video supplement, identified anomalies can manifest in various shapes. Anomalies are typically located in valleys, folds, or generally along depressions of the tropopause boundary. Because the algorithm works in a broader sense to identify irregularities along the boundary, it does not distinguish between the type of anomaly. Especially for our case study, we use the East Asian region shown in Fig. 9b to analyze the event.*"

After this visual introduction, the more technical part follows about the algorithm itself. We put effort in creating the link to the already introduced 2-D description, using the same attributes and names, also referencing the 2-D identification itself. This paragraph starts by revising the distance measure used in 2-D and motivating the application in 3-D using a scale analysis. The distance measure is clarified before the description of the algorithm.

Then, we explain the 3-D identification along a newly created figure (see below), showing the workings of the algorithm using a vertical cross section. We give a general overview over the attributes and names. Color maps and attributes have been chosen to match the already introduced 2-D overview figure. We refer to the 2-D overview figure (old Fig. 4) to create a clear transition from 2-D to 3-D.

[revised manuscript text omitted]

After the technical description, we go over the filtering process, which has been kept primarily as in the first submission, but with some clarifications based on the reviewers' comments ("Line 389-390"). We kindly refer the reader to that specific comment. We finalize this section with a short explanation on the question whether existing PVS identification techniques could be applied to 3-D data.

"Existing identification techniques (see an overview of them in Papin et al., 2020) rely on following the 2-PVU boundary, and searching points along this boundary meeting certain criteria (e.g., Sprenger et al., 2007). However, in 3-D, there is no unique direction to follow the boundary (the dynamical tropopause), making such techniques not applicable. Our strategy does not rely on a spatial tracking of a contour, but works in a broader sense, where the introduced domain $\Omega$ and boundaries $\Gamma_1$, $\Gamma_2$ can be extended to a 3-D perspective."

**Specific Comments**

Line 23: May want to list some of these nonconservative processes. Also, what do you mean by "large-scale"?

We agree that some extension of the presentation at this point improves the Introduction. We used the term "large-scale" as a rather muddy surrogate for "in the absence of strong latent heat release". We have clarified this point and the extended and revised version now reads:

"A key component of this framework is that PV is materially conserved in the absence of nonconservative processes such as, e.g., latent heat release in clouds, longwave radiative cooling, or turbulent mixing. In the absence of strong latent heat release, it turns out that nonconservative PV modification is comparatively small and that material conservation of PV provides a very good first approximation. When considered on isentropic levels, the temporal evolution of PV at a given location is hence governed by quasi-horizontal advection only, which typically yields a rather smooth PV evolution." (Lines 22-25)

Lines 163-165: How do you merge the data at the equator? Is it possible to get different results from the two mappings? Or do you just ignore this region for this study and leave the boundary problem for future work?

We agree with the reviewer that there is room for misunderstanding in our initial presentation. The data is not merged along the equator. Instead, as described in Lines 163-166, two distinct stereographic projections are generated, one for the northern and one for the southern hemisphere. The only difference between these two distinct areas is the sign of the PV. When considering global data, the resulting PVAs of both hemispheres can be appended. Clearly, with this strategy anomalies in the equator region cannot be robustly identified.
However, in this study we focus on the identification of anomalies along the dynamical tropopause, which lacks a proper definition in this area (as mentioned by Reviewer 2).

Therefore, regarding the aim of this study, anomalies in the equator region are being not considered. Nevertheless, it is noted that the algorithm can be used in a more general sense to identify any anomalies relative to a main reservoir. Future work can apply this versatile algorithm to different domains or even different fields. In case the user is interested in anomalies near the equator, the hemispheric domains can be expanded towards the opposing pole, e.g., use for the northern projection an area from 20°S to 90°N. This also works because we compensate for distortions in the projection.

We have clarified the interest of the present study is not in the area around the equator: *"The singularity at the pole shifts to a boundary at the equator. Since the dynamical tropopause changes its sign at the equator and generally lacks proper definition at lower latitudes, this area is not considered as a region of interest for this present study."* (Lines 165-166)

Furthermore, we agree with both reviewers that the abstract and the publication should clarify the use of the algorithm in this study (restriction on the dynamical tropopause). The outlook may then state the versatile nature of the algorithm and possible future extensions. Text changes made are noted in Comment 2 of Reviewer 2.

Line 173: Is this angle measured with respect to the center point (xc, yc)?

In this line, we do not refer to any angle explicitly. It is thus not completely clear to us what the reviewer means to ask here. In general, angles used in this section are relative to the center of the projection, which is the pole N in Fig. 3 or indeed the point $(x_c, y_c)$. The angle relative to the projection center is introduced at Line 178.

Line 177: Should the exponent be 2 rather than -2. Otherwise, what does cos-2( ) represent?

In this case, $\cos^{-2}(x)$ denotes the inverse of the cosine-squared function, so $1 / \cos(x)^2$. We agree that the current version might create confusion to the inverse function $\arccos(x) = \cos^{-1}(x)$. Therefore, we moved the exponent -2 to the end of the equation.
In TeX notation (Line 177): h=\cos(\frac{\theta(x,y)} {2})^{-2}

Line 184: I don't understand how $\Delta s(x_c, y_c)$ is calculated. Can you give an example or show this distance on Figure 3?

We agree to both reviewers that this step requires more clarity than in our initial submission. $\Delta s$ for a single pixel in the projection can also be computed by taking the spherical distance from this pixel to one of the neighboring pixels. Due to the conformal property, all neighboring pixels yield the same result (apart from discretization errors). This spherical distance is computed for the pixel at the projection center $\Delta s(x_c, y_c)$, and used as a reference point to compute the entire field $\Delta s$. This calculation is not feasible to include in the Figure, therefore we included a more distinct explanation:

*"$\Delta s(x_{c},y_{c})$ denotes the radius on the sphere that projects onto a circle on the projection plane centered around (x_{c},y_{c}) with a radius of one pixel. Since the pole is the projection center, the pole of the sphere touches the projection plane in (x_{c},y_{c}). By mapping one neighboring pixel of the pole to their regarding latitude and longitude coordinates, $\Delta s(x_{c},y_{c})$ can be obtained by computing the spherical distance between the pole and this neighboring pixel. Due to the conformal property of the projection, any of the neighboring pixels yield the same distance."* (Line 184)

Line 196: Does z represent an infinitesimal line segment between any two arbitrary points?

Yes, this is correct. We integrate over infinitesimal small line segments from the boundary to the destination point x, and therefore we can evaluate the cost function ∆s at any given point in the field. Because this path integral is vital for the algorithm and we agree that the definition should be rephrased, we changed the section in question as followed:

*"We search the shortest distance u emerging from any point in the boundary Γ to x with respect to a given cost function $\Delta s$:*
*u(x)=\min_{\gamma\in\Gamma}\int_{\gamma}^{x}\Delta s(z)dz"* (Lines 194-196)

Line 201: What is the gradient used here? Is it a spatial gradient along the projection plane?

Yes, ||u|| is the spatial gradient in the distance field along the projection plane. We rephrased Lines 205-206 to give the reader a better understanding of the in the equation introduced gradient:

*"The field of shortest distances u along the projection plane contains the distances from a given boundary to points in this field. Intuitively, the gradient of u can be thought of a measure that is anti-proportional to the distortions defined by the distance field $\Delta s$."* (Lines 205-206)

Figure 4 caption: I know this is an illustrative figure, but it would be helpful to know what dataset, time, vertical level, as in Figure 5, in case a reader may want to try to reproduce these results.

We agree and add a sentence to the caption containing sufficient information to reproduce the results for the reader:

*"Visualized is the 335 K isentrope from the ERA5 reanalysis at 7 September 2016, 00 UTC."*

Lines 240-1: The threshold PV value (2 PVU) could possibly be classified as an input parameter. You don't show any sensitivity to the choice of this parameter, but clearly it could have a significant impact on your analysis. You may want to comment on this.

We did not perform any sensitivity analyses regarding the PV domain. This comment falls in line with the "Line 163-165" comment, as well as the 2nd comment by Reviewer 2, in the sense that we agree the need for a clarification regarding the focus of our study. Our work focuses on the identification of anomalies only along the dynamical tropopause. We rephrased parts of the abstract and introduction in that manner and kindly refer the reviewer to the 2nd comment by Reviewer 2 for the changes to the text. Nonetheless, the algorithm is versatile. On one hand, it can be applied in a broader sense to detect anomalies attached to a broader reservoir, and on the other hand, it indeed allows sensitivity testing. In this study, however, the methodology and this versatile nature of the algorithm are the essence, but not studying the sensitivities of using different thresholds.

Line 255: What about features that are disconnected from the main part of the region, like the small filaments in the Atlantic Ocean. How do you calculate their distances since no path from the inner boundary is available that stays within the domain?

This is correct, the algorithm cannot reach cutoffs since they are not connected to the main stratospheric reservoir. We remark this fact in the algorithm by setting their distance from the reservoir to infinity. This makes them also clearly distinguishable from attached anomalies later in the pipeline. We agree that this information should be added to the text.

*"Keeping this domain is necessary to measure distances following the stratospheric domain with respect to the distance map (as illustrated in Fig. 2). The distances can be seen in Fig. 4f. While filament-like structures are assigned meaningful distances, note that cutoffs cannot be reached by the algorithm in this step because they are not spatially connected to the stratospheric reservoir. Their distance is set to infinity, making them easily recognizable later. Feature descriptions for these objects can be computed, nonetheless."* (Lines 256-257)

Line 254: Is the contour here defined as the boundary of the object? If the object is connected to the main reservoir, do you include the line segments where the object and the reservoir intersect?

The contour $\Gamma_2$ is the boundary of the inner stratospheric core (black line in Fig. 4e). This has been clarified in the text passage as followed:

*"The outer contour of the previously defined inner core (red/black contour in Fig. 4d/e) is defined as boundary $\Gamma_2$. This boundary is the starting point for the next operation."* (Line 252)

Areas further than w/2 km away from this inner core are marked as anomalies (grey in Fig. 4g). The line segments (orange in this panel) are not computed explicitly since they are not required, they are just implicitly defined by the boundary of the anomalies. Of course, it is possible to compute these as well, serving as an additional metric for the object descriptions.

Section 4.4: This is a nice description of various metrics, but it seems that they are not really used much in the paper. It would be nice to have some examples of these calculations for different features in one of the 2-D examples.

Most of the descriptions are used in Figures or examples within the paper, but we agree with the reviewer that some of these should be elaborated and reference better in this section. Therefore, we added text along with the associated feature descriptions to emphasize the usage of these.

Object area: Added sentence (Line 280):
*"Besides as filtering metric, the object area is a component of some of the subsequent mentioned metrics."*

Length: Modified sentences (Line 284):
*"Our identification strategy yields a useful and robust length measure, while other identification strategies only measure the length along the 2-PVU contour. The length is used as a filtering parameter, as further elaborated in Sect. 5, and well suited for statistical analyses."*

Centroid: Added sentences (Line 289):
*"Centroids give a quantifiable estimate of the object's center. For example, in Fig. 6b the centroid of an object lies in the intersection of the green main axes. Their tracks can be used to investigate case studies or big data analyses. In the 3-D case, it also reveals the vertical movement of the anomaly, as outlined in the case study later on."*

Figure 6: It is interesting that this figure is based on pressure layer-averaging rather than isentropic level. Is there a reason for this? Also, it is unclear how the individual objects are distinguished here. For example, the feature over the Atlantic Ocean appears to have two distinct PV anomalies. Are they both part of one object?

With respect to your first question, this data set has been provided and analyzed by one of the Co-Authors in Maier-Gerber et al. (2019, 2021), where we investigated cases of a Tropical Transition with the involvement of PV. Generally, data is often not available on isentropic levels, then the PV on (averaged) pressure levels pose an alternative for analyses. Also, pressure level data can be handled and interpreted differently: While the height of an isentrope highly depends on the latitude, pressure levels usually are roughly at the same height. All in all, this should also show the versatility of the algorithm in the 2-D case.

Moreover, we agree with the reviewer that the second question might arise to the reader and has not been discussed in the original version. Yes, from a meteorological standpoint these two anomalies have a separated history. Since the anomalies indeed touch, this specific case considers the structures as merged and therefore as a single anomaly.

However, deciding which object consists of multiple distinct (yet connected) anomalies states a problem that gets highly complex when considering merging and splitting of these anomalies over time. More sophisticated heuristics are required to decide on their distinctness, e.g., tracking their origin or trying to split features which are loosely connected. Still, note that other identification strategies are not identifying an anomaly in this region at all.

We clarified this fact by extending the paragraph with the following sentences:

*"In this specific case, the two anomalies are spatially connected. Our strategy therefore considers them as one merged object. Further heuristics are required to distinguish the origin of these merged anomalies in order to separate them, but this issue is beyond the scope of this work."* (Line 324)

Maier-Gerber, M., Riemer, M., Fink, A. H., Knippertz, P., Di Muzio, E., and McTaggart-Cowan, R. (2019). Tropical Transition of Hurricane Chris (2012) over the North Atlantic Ocean: A Multiscale Investigation of Predictability. *Monthly Weather Review* 147, 3, 951-970, available from: https://doi.org/10.1175/MWR-D-18-0188.1 [Accessed 04 April 2022]

Maier-Gerber, M., Fink, A. H., Riemer, M., Schoemer, E., Fischer, C., and Schulz, B. (2021). Statistical–Dynamical Forecasting of Subseasonal North Atlantic Tropical Cyclone Occurrence. *Weather and Forecasting* 36, 6, 2127-2142, available from: https://doi.org/10.1175/WAF-D-21-0020.1 [Accessed 04 April 2022]

Lines 327-328: Regarding the length calculation, is this only for features that are connected to the main reservoir? What about the cutoff features in the ocean? How do you calculate the length of those features for the filtering algorithm to use, since they are disconnected? This is related to an earlier question.

See our answer to the comment regarding Line 255.

Lines 320-323: I don't understand this sentence. How does their algorithm identify a broad trough? Are your referring to the 2 PVU contour on Figure 8b? And what is the tropospheric air encapsulated in the stratospheric reservoir? Can you point to that in the figure?

The main difference of our algorithm to existing ones revolves around the fact, that our algorithm makes decisions based on the information of the entire tropopause, while other strategies only take the outermost 2-PVU boundary into account. In this example, we are indeed referring to the 2-PVU contour on Figure 8b. We added a red arrow at the region of interest in the Figure (see below) to emphasize this outermost contour. This should make it clearer that other strategies, which only follow the red arrow, are not able to take the inner state of the PV reservoir into account, e.g. tropospheric cutoffs (tropospheric air which is fully surrounded by stratospheric air).

We changed the text at Lines 320-324 as followed to state more clearly what the algorithm identifies and reference the changed figure better from therein:

*"These structures are not identified by the approach from WS07. By only tracing the outermost 2-PVU boundary (red arrow in Fig. 8b), the inner state of the PV domain is not considered. However, this inner state might reveal important information on the large-scale flow in this area. By just following this red arrow, this results in a broad PV trough being spotted from the Central Atlantic all the way to the Black Sea. This wide trough does not fulfill the required thresholds for a streamer. The strategy we employ also considers the inner state of the reservoir, including all troposphere-stratosphere boundaries in the domain. The erosion step starts at these inner boundaries as well. Therefore, these elongated structures are not reconstructed in the following dilation step, identifying them thereafter."* (Lines 320-324)

[Figure]

Revised caption for Figure 8:
*"Same as Fig. 7, but with the data set used in Fig. 6. The red arrow indicates the outermost 2-PVU contour, which is used to identify anomalies in the work by Wernli and Sprenger (2007)."*

Line 367: The first sentence seems to be missing some words "extend the in Section 4"

To make this sentence easier to read, we rephrased it to:
*"There are several possibilities to extend the algorithm introduced in Sect. 4 to 3-D."* (Line 357)

Line 370-379: The description of the 3-D extension of the algorithm seems to be lacking important information for the reader. Here are a few questions and comments.

We thank the reviewer for their comments on this part of the paper. This is an integral part since it introduces the novel concept in 3-D and should be distinct and clear to the reader. However, we agree based on the comments, that this is currently not the case. We did major changes and added a new figure to this section. Details are given in the General Comment above. We want to address the comments by the reviewer, nonetheless.

Clarifications in the paper regarding these comments have been incorporated in the text changes outlined in the General Comment.

• In 2-D you have boundary defined by the 2 PVU contour and domain defined by the area enclosed by that contour, which includes PV values greater than 2 PVU. In 3-D the boundary is now the 2 PVU surface, but what is the domain? Is it the 3-D region where the PV value is greater than 2 PVU? So generally you are calculating distance upwards from the 2 PVU surface?

Yes, this is correct. The definition of the domain is the same in 2-D as in 3-D. It is defined by the area exceeding the 2-PVU threshold. Therefore, distances are calculated upwards, but not JUST upwards. They are calculated in all directions in 3-D space. The new panels introduced in the General Comment give a better overview over the definitions, as well as the refined text.

• In 2-D you calculate the distances constrained by lines that remain in the domain. But in 3-D are the distances still required to stay in the domain? How is that done?

That is exactly the elegant peculiarity of the algorithm: All operators work in 3-D in the same fashion as in 2-D, but with an added dimension. Therefore, while in 2-D we measure distances constrained by the boundary lines (2-PVU isoline), in 3-D we measure distances constrained by boundary surfaces (2-PVU isosurface). Starting at these boundaries, the employed algorithm can measure distances in 3-D space rather than 2-D space.
We hope to clarify these ideas as well in the refined text and added figure.

• This 3-D calculation takes into account both vertical and horizontal displacements, and in the vertical you use a scaling between potential temperature and height. Couldn't you use the geopotential height as the approximate vertical position of each grid point?

During the writing process of this paper, we took different ideas into account regarding the vertical dimension and scale analysis. We concluded that the current version is the most sensible trade-off between complexity and interpretability. As introduced in Line 371, we also took the idea into account to use Euclidean distances in the vertical dimension, based on a standard atmosphere model. However, this increases complexity strongly. Furthermore, potential temperature increases by definition with height regarding isentropic levels trivially. Other fields, like the suggested geopotential height, could include inversions in the vertical domain, making distance measures very complex to handle.

• Are there complications in your algorithm when there are multiple tropopauses in one vertical column? How does the algorithm handle these?

The algorithm does not distinguish between the types of anomalies. Tropopause folds (what the reviewer is refering to) are just one of the possible manifestations of an anomaly along the dynamical tropopause. The algorithm searches anomalies along this tropopause, therefore folds are identified as well. The introduced new figure gives an example in 3-D of an anomaly which has partly multiple tropopause crossings in a vertical column.

• It could be that one additional figure showing how the process works in 3-D with a simpler shape than in Figures 9 and 10 would help the reader to understand this extension. Could you make plots similar to Figure 4, but with the 3-D domain? For 3-D, it is not obvious what is the domain, the two boundaries, the direction for which the distances are calculated, etc.

We fully agree and think an additional figure might help the reader to understand the step from the 2-D to the 3-D identification. Especially the counterpart of the variables in 2-D to 3-D seems to be unclear and how the line boundaries are surfaces in the 3-D case.

Figure 9: It looks like the anomalies are above the dynamical tropopause, extending into the higher PV stratosphere. Is this an optical illusion? Also, I wondered why the long streamer in the central lower part of Figure 9a is not identified as an anomaly. Is this too thin in the vertical to meet the 6 K threshold?

Anomalies are expected to "sit" right on the dynamical tropopause, mostly in "valleys", "folds" or generally along depressions. We put a lot of effort in creating clear visualizations, however, 3-D visualizations on a 2-D piece of paper are naturally difficult. Some anomalies reaching a bit further into the stratosphere than expected, this is a result of the different types of fields that have been merged in this display. The colored anomalies are a smoothed binary field, while the PV field itself is continuous and differentiable. This leads to isocontours from the data sources being naturally a bit off track. The video as supplement should also give the reader more insight and a clearer depiction of these anomalies.
Especially towards lower latitudes, south of the jet stream, most of the anomalies are very flat. These anomalies are often not fulfilling the threshold, one of these anomalies is the one mentioned by the reviewer. The most vertical extended anomalies are along the jet stream.

Line 382: Would this be analogous to $\Gamma_1$ in Figure 4a? What would $\Gamma_2$ look like?

We agree with the reviewer that this requires more clarity: At Line 382 we used $\Gamma$, but in the 2-D introduction we used $\Gamma_1$ and $\Gamma_2$ instead. We are clarifying what $\Gamma_1$ and $\Gamma_2$ are representing in the 3-D case, see the General Comment and the new panels for details.

Line 389-390: Centroid and area don't seem to be calculated anywhere. Can you give an example?

Formulas for the computation of centroid and area have been introduced in Sect. 4.4, and they are applied in the case study. We added a reference to the equation in the text as followed:

*"The centroid of an anomaly is computed in 3-D (see Eq. 8), while we replace the area measure with a volume measure (in km²K, or approximated in km³ using a standard atmosphere model). Using the centroid in 3-D, the vertical evolution of an anomaly can be analyzed. The case study later investigates an anomaly with a clear shift in vertical position over time."* (Lines 389-390)

Line 400: This says you filter the areas to get detached and separated objects. It seems arbitrary that the 6K cutoff would separate the regions into dynamically distinct features. Won't this artificially cut off certain features that should be connected?

The threshold of 6K was chosen empirically, as shown in Fig. 10, but heavily supported by 3-D visualizations we conducted. Any set threshold will separate dynamically connected features. Looking at the 3-D data field, it becomes clear that all anomalies around the globe are connected in some "circular fluid-dynamic pot". But this would not yield to a set of identifiable objects. Therefore, we agree with the statement by the reviewer that this artificially cuts off dynamically connected features. This is a drawback of our goal to have features which can be individually attributed. This approach is explained at Lines 395-399.

Line 469-470: The yellow anomaly is an equatorward intrusion of stratospheric air. Since tropopause height increases equatorward, wouldn't the anomaly lie under the tropopause, like a tropopause fold?

Tropopause folds lie within regions of multiple tropopause intersections in a vertical column, yet they are connected to the stratospheric reservoir. In a more general sense, the algorithm detects anomalies along a given boundary, here the tropopause. Therefore, it does not distinguish between the types of anomalies: They could be folds, typical "streamers", structures in-between or something completely different like "stalactites"; all of these are detected. Objectively, identified folds can be defined as a subset of the identified anomalies, ones, which have at least partly a layer of tropospheric air above them and below them.

We introduced some types of anomalies in the 3-D description, which is elaborated in the General Comment. We state that folds are also a type of anomaly along the dynamical tropopause, hence they are identified.

Figure 13 gives the appearance that the anomaly is above the tropopause. Maybe it would help to explain exactly what the isosurfaces of the anomalies represent in your 3-D diagrams.

Based on the remarks and questions by both reviewers, we agree that there is a need for a crisper and clearer introduction of the step from the 2-D to the 3-D identification, especially regarding the in the 2-D case introduced parameters ($\Gamma$,$\Omega$). The position of the

anomalies has been clarified in the Comment regarding Figure 9. More details regarding our changes to this section are described in the General Comment.

Technical Corrections
Figure 2 caption: Do you mean "FMM" here, not "FFM". Also, do you want to capitalize "marching method"?
Line 246: replace "is" with "are", since the noun is plural (distances)
Line 281: Do you mean distance d emerging "from" ...
Line 305: Should "form" be "forms", since the noun is singular (shape)
Line 323: "spot" should be "spots"

We thank the reviewer for pointing out these technical corrections. These will be incorporated into the paper where these corrections are still necessary.

**Reviewer # 2:**

Specific Comments

The word anomaly can be misleading. In general, it is easy to understand what it means by the context, but not as much in the abstract.

We agree that, without further context, the term "anomaly" may leave room for misunderstanding. In the abstract, where further context is not given, we have thus substituted this term with "feature" (Lines 2, 3, 5), which is consistent with our use of that term in the main text. The second paragraph in the introduction discusses examples of the type of anomalies that we consider in this study. The reviewer seems to agree that our use of the term "anomaly" is sufficiently clear in the main text. We thus did not modify the text elsewhere.

Abstract: "The generated feature descriptions are well suited ... for generation of climatologies of feature characteristics". Lines 534 to 536: "Further research is required on certain aspects of this identification ... for climatologies, automated tracking techniques are required. Ideas for further work ..." I do not think the 3D algorithm is that well suited for climatologies in its current version, feature descriptions or not.

We agree with the reviewer that the abstract indeed suggests that climatologies are performed within our work, which is not the case. The purpose of this formulation is to show up the potential and possibilities of the algorithm. From a technical standpoint the algorithm is implemented into a framework which is highly parallelized and suited for big data analyses. Furthermore, having a seasonal independency regarding the configuration is one of the major advantages of the new strategy in 3-D. Therefore, climatologies over

the entire season can be computed without having to choose an appropriate isentrope. The mentioned tracking would be an optional component, which can give further insight into the identified structures.

Therefore, in the abstract we remove the claim that we compute climatologies:

*"The generated feature descriptions are well suited for subsequent case studies of 3-D atmospheric dynamics as represented by the underlying numerical simulation"* (Lines 10-12)

And at the end of the abstract, after listing advantages, we mention that these advantages open applications for the computation of climatologies:

*"These advantages, as well as the suitability of the implementation to process big data sets, also open applications for climatological analyses. The method is made available as open-source for straightforward use by the atmospheric community."* (Line 18)

Looks like the development and testing of this method was restricted to the detection of anomalies in the dynamical tropopause. However, both the title and the abstract suggest a more general use. This is not discussed anywhere. There is just a mention of the problem in the equator due to the projection (lines 165-166), but this doesn't affect the dynamical tropopause, which is not defined at the lower latitudes. Can the method be used in other heights (i.e., middle troposphere/stratosphere) as it is?

We agree with both reviewers that the paper requires a clarification that our identification strategy focuses on anomalies along the dynamical tropopause. Nonetheless, the algorithm can be applied in a broader sense to detect anomalies attached to a broader reservoir. The current abstract suggests a more general use. In the authors' opinion, a clarification in the abstract should be enough without changing the title of the publication.

The abstract gets changed as followed:
*"This study presents a novel algorithm for the objective identification of PV anomalies along the dynamical tropopause in gridded data, as commonly output by numerical simulation models."* (Lines 5-6)

The definition of the dynamical tropopause is already present in the text and requires no further elaboration in the abstract. Also, as stated in Review 1 (Line 240), we should mention in the outlook the versatile nature of the algorithm. While we apply it in the study to the dynamical tropopause, the algorithm can be used in a broader sense to identify anomalies along a 2-D or 3-D boundary, not necessarily restricted to PV fields.
The first sentence of the conclusion has been adjusted to also indicate what we focused on in this study.
*"In this study, a novel automated identification strategy for PVAs along the dynamical tropopause has been presented."* (Line 486)

Then, at Line 533 we extend the paragraph to include the following:
*"Although this study focuses on the identification on anomalies along the dynamical*

*tropopause, the algorithm in a broader sense can be applied to identify anomalies along any 2-D or 3-D spatial boundary. Therefore, other identification tasks in meteorology can also benefit from these ideas and strategies."*

Line 184: "$s(x_c, y_c)$ itself can be calculated from the latitude and longitude positions of the pole and its surrounding pixels" This should be explained.

We agree with both reviewers that this part requires more clarity. The answer and corrections have been given above (Reviewer 1, Comment "Line 184").

I have had a lot of problems with "visualizing" the 3D method. The representation of the PVA in Figure 11 b and c (for example) is confusing: it looks like the 3D surface is such that it encompasses a volume (and then $\Gamma_1$ and $\Gamma_2$ should be surfaces, too). But this is not possible if the represented surface is 2 PVU: how is the surface closed to make it look like the PVA represented in Figure 11c?. It took me some time to realize that the computed surface doesn't have a thickness, and thus $\Gamma_1$ and $\Gamma_2$ are lines. Even in the images in the left column of Figure 12, where the structure of the surface is better represented, it looks like the anomaly has a thickness, a volume. In summary: explain what you do to make the PVA look like that (and why) or use a more realistic surface.

We understand that both reviewers and therefore also potential readers have trouble understanding the step from the 2-D identification to the 3-D identification and are thankful for pointing out the difficulties in this regard. Both $\Gamma_1$ and $\Gamma_2$ are surfaces indeed.  $\Gamma_1$ represents the dynamical tropopause (2-PVU boundary). This is the greenish isosurface in the 3D visualizations. It does not have to be closed; it just needs a well-defined boundary. $\Gamma_2$ similarly is also a surface, representing the boundary of the shrunken reservoir. The anomalies themselves are then 3-D objects encapsulating the volume of the anomaly. Typically, these are 3-D areas of valleys or depressions along the tropopause.

We addressed this further in our response to the general comment of Reviewer 1. There, we give further explanations, show the changes to the text we made, and introduce a new figure. Especially the panels of this new figure should help the reader to take the step from 2-D to 3-D by using similar notations ($\Gamma_1$ and $\Gamma_2$) and similar color schemes as Fig. 4 (2-D schematic). We kindly refer the Reviewer for details to this response.

In the left column of Figure 12, it is difficult to see the yellow region as an anomaly, and the right column does not help much. It would be useful to see two cross-sections going through the anomaly (a horizontal one and a vertical one), at least for one of the rows. I know this means losing information but, if the sections are wisely chosen, it will also mean gaining perspective and understanding.

We also address this issue in our response to the general comment of Reviewer 1. There, we introduce new panels featuring vertical cross sections. We hope that this figure will help the reader to get a clear understanding on the 3-D structures of the field and the

anomalies. Furthermore, we hope the video in the supplement helps the reader to improve the understanding of the 3-D visualization. It contains an animation of the left and right column of Figure 12. The moving camera should help to get a clearer understanding on the shape of such anomalies. We kindly refer the Reviewer to the General Comment for the revised text and new figure.

---

## Author Response (AR2)

**REVISED SUBMISSION: RESPONSE TO REVIEWERS A novel method for objective identification of 3-D potential vorticity anomalies 10.5194/gmd-2021-424**

We would like to thank the reviewers and the handling topical editor for further comments and feedback on our manuscript. Detailed point-by-point replies (blue font color) to each comment are provided below. Technically, citations of text passages are in italics. Changes made to the text are highlighted in yellow. Minor changes to some figures are described here as well, however, these figures are not included in this answer. Line numbers refer to the resubmitted manuscript.

**Handling topical editor:**

**Dear authors,**

The reviewers have recommended minor revisions and acceptance of your work, respectively. Please, check the comments by the Reviewer 1.

Also, I have read with attention your manuscript, and I think that the reviewers have done a great work reviewing it. I think that using a stereographic projection is a very clever idea. My interest comes from the fact that the identification of PV fields is a topic that I researched a few years ago, also related to identifying the tropopause. You can consult it here: https://doi.org/10.1371/journal.pone.0072970

The obligations for editors of GMD state that "Editors themselves should be extra careful in suggesting additional literature." Our publication policy also focuses on avoiding citation malpractice. However, given the topic, I think it is justified to point out my work in this case.

Beyond the PV streamers, in line 38, you mention "cross-equatorial" intrusions. We also found this problem with our algorithm to detect PV fields. In several cases, this happened, and we had to consider it. Somehow, it is similar to the problem of using a longitudinal cartesian projection and dealing with a PV surface cut at a given meridian. We solved it simply by computing surfaces separately and aggregating areas. Our problem was a 2D surface, so simpler than yours.

Also, in lines 145-149, you talk about the discrete nature of algorithms and their impact on the computation. Somehow, this was precisely the problem we addressed with the ROI algorithm instead of piecewise-constant techniques. The improved interpolation and granularity behind the ROI implementation in IDL let us improve the accuracy of the computation of the PV surfaces, independently of the initial horizontal grid, with an increase in precision up to ten times and a computing time up to 9 times faster for a 1,5 degrees grid.

I think that it could be fair and that the point made in your work would benefit from citing our previous results on these two issues, as they show that similar problems have been faced in the past, and that there is room to improve the solutions when computing this kind of structures.

However, I want to clarify that one referee now recommends accepting your paper as is. Another requests only a few minor modifications and does not consider it necessary to review your work again before acceptance. Therefore, independently of your decision on citing or not the work that I point out, your manuscript is almost sure to be accepted for publication if you address the recommendations of Reviewer 1.

Regards, Juan A. Añel Geosci. Model Dev. Executive Editor

We thank the handling topical editor for his remark and for leading the reviewing process. The comments by Reviewer #1 have been addressed. Our responses to these comments are provided below.

The work mentioned by the handling editor uses a region of interest (ROI) approach to significantly improve area measurements, which results in a more precise computation of the equivalent latitude. On the other hand, our work requires a robust measure for distances in a projection following a given field. While the applications are quite distinct, both approaches use higher-order schemes in numerical computation to compensate for distortions, relating these works to each other to some extent.

Regarding the first text passage mentioned by the handling editor (Line 38): We avoid problems at the anti-meridian specifically by choosing a different projection. Furthermore, we are here not specifically interested in areas near the equator. The choice of this specific projection has been extensively discussed in Lines 150-170. We thus do not consider a reference to your previous work at this point to be very helpful.

However, regarding the second text passage (Lines 145-149), we are indeed faced with a similar problem to the one discussed in previous work by the handling editor. While the applications and the solutions are different, the fundamental problem are closely related from the computer-science perspective. Therefore, we agree that our paper benefits from a reference to these previous results and we changed the text as followed:

"Using this concept in real-world environments unfortunately is non-trivial. The distance measure required must follow the domain instead of using a direct spherical distance, as seen in Fig. 2. Most algorithms that satisfy this requirement suffer from metrical errors induced by a discrete representation of the projected grid. Instead, Añel et al. (2013) achieve significant improvements in area computations using a higher-order numerical scheme based on a region-of-interest approach. On the other hand, the present study requires an approach to compute distances (as shown in Fig. 2b), but similarly must compensate for distortions in the field. For more precise results we choose a strategy based on a higher-order numerical scheme also, as outlined in the next section." (Lines 143-149)

Añel, J. A., Allen, D. R., Sáenz, G., Gimeno, L., and de la Torre, L.: Equivalent latitude computation using regions of interest (ROI), Plos one, 8(9), e72970, https://doi.org/10.1371/journal.pone.0072970, 2013.

**Reviewer #1:**

**General Comments**

I appreciate the author's revision and response to both sets of reviewer questions. I believe the paper is nearly ready for publication. Just a few minor comments and suggested technical corrections are listed below. Figure 10 was a very helpful addition to the paper. As I read this, I wondered whether the algorithm could be extended to 4-D, taking the time-evolution into account? Not sure how the "distance" would be calculated in that case, and how exactly the boundaries would be defined. Probably too complicated, but it would be interesting to know whether one could track individual PV anomalies as they flow through time.

This is a very interesting question and remark. Regarding tracking of PV anomalies, we performed some experiments mainly on the 2-D case, however, not adding the time dimension as third dimension to the algorithm but using spatial overlap heuristics and distance measures in the feature vector space to find corresponding anomalies of subsequent timesteps.

In theory, the presented algorithm could also be applied to 4-D, but this stresses the authors' limits of visual thinking. The 2-D and 3-D strategy handles "intrusions in the spatial dimension". These can be clearly visualized and thought of, as described in the paper. The fourth dimension would add "intrusions in time". First, note that identified objects are labeled by coherence. Therefore, the tracking can essentially be thought of a form of overlap tracking. However, to decide which areas to consider as anomalies, the operators (dilation and erosion in the time dimension) would also highlight structures that are "short-lived" (c.f. thin structures in the spatial domain  $\cong$  short-lived anomalies in areas which are broad and not spatially covered by the anomalies. It is questionable, yet debatable, if such structures are of interest, and it opens a variety of questions regarding the distance measure and the filtering process.

All in all, considering time as fourth dimension can be useful, and it can combine identification and tracking in one cohesive process. For our specific use case, the extension of the used operators to 4-D is possible, but would become very complicated, difficult to interpret, and would lead to more questions than it would solve. Not applying the dilation and erosion operators along the time dimension would yield a simple overlap tracking, which might already give good results if the temporal resolution is chosen wisely. We hence refrain from mentioning a possible generalization to 4-D in this study, but keep in mind this interesting thought for future work.

**Specific Comments**

Lines 127-8: In the erosion exercise in Fig. 1, it looks like the mask is small enough that there would be some part of the extended PV intrusion that would be kept and should show up as a greyed out area in panel (b). Is this true, and if so does the algorithm remove these type of cutoffs during the erosion? Later, lines 261-3, you say "Generally, this inner core may contain multiple disjunct areas. In these cases we pick the biggest one as the core to proceed with, defined by its area." I assume a similar process is done for this figure. If so, you may want to state that explicitly to avoid any confusion. It might actually be helpful if you greyed out the smaller inner core area (if it does indeed exist) to illustrate this.

The figure in question has been generated in a manner that the orange mask should be big enough to not result in any left-over greyed area within the anomaly. The idea of this section is to provide a quick introduction to the operators used in the algorithm while sticking to a simple example. As noted by the reviewer, it is essential to pick only the biggest "inner core" after the erosion. We decided to not include such a specific case in this introduction.

However, after reviewing the figure, we agree that the orange mask in Fig. 1a-c does not quite cover the entirety of the displayed anomaly. The size of the displayed orange mask has been adjusted, and the figure has been revised accordingly to match the actual used size.

Fig. 4h: This is a very minor point, but I'm not sure what the green and red symbols represent. Are they supposed to be funnels? Maybe a green check mark and a red X would make more sense to the reader?

Yes, these funnels should symbolize the filtering process. However, we agree that a check mark and a X would be more intuitive and would also be the better choice regarding the color accessibility guidelines. Therefore, the figure has been revised by changes the funnels to green check marks and red crosses.

Page 19: When I first read the revision, I wondered why the paragraph starting "Figure 9a..." had been moved from its previous position. In the original submission it was placed after the description of the 3-D algorithm. Now it is before the description, and the new Figure 10 is referred to in the algorithm description. After reading this section several times, I think I understand that you're trying to orient the reader to the 3-D rendering of the tropopause with this figure. If so, would it make sense to not highlight the PV anomalies in Figure 9, but just show the tropopause, as in Figure 4c? Then later, after describing the algorithm, you could show this figure again with the anomalies identified, as in Figure 4g? Just a suggestion, because it is a bit confusing to see the anomalies in this figure, but then later explain how the anomalies are identified.

We thank the reviewer for this comment. With Figure 9, we want to give the reader an overview and a visual introduction to both the entirety of the tropopause in 3-D, but also the structure, distinctness, and locations of the anomalies. In the mentioned section, we first introduce the tropopause displayed in the figure, then we show how the anomalies are situated in relation to the tropopause. We think doubling this figure, one with and one without anomalies, is not necessary and that it is sufficiently clear in this case.

Is Figure 9 for 24 July 2015, as in Figure 10? May want to include the date/time in the caption.

We thank the reviewer for pointing out the missing reference on how to reproduce this display. Indeed, the same time step has been chosen. We added at the end of the caption of Fig. 9: *"Displayed on both panels is the ERA5 reanalysis at 24 July 2015, 00 UTC."*

Line 467: Was Met.3D used for the previous visualizations? If so, you might want to move this statement above to where you first use this software (Figure 9 description).

Yes, Met.3D has also been used for the previous visualizations. Although we already introduced the tool in the introduction (Lines 105-111), we agree that there should be a further remark when referencing the first figure making use of Met.3D instead of the second one. Therefore, we removed the reference to Met.3D in Line 468 and added: *"Figure 9a shows a 3-D visualization using the open-source visualization framework Met.3D (Rautenhaus et al., 2015b)* of the tropopause (defined by the 2-PVU isosurface) and the identified anomalies." (Lines 388-389)

**Suggested Technical Corrections**

These are mainly stylistic suggestions that you can take or leave as you see fit.

Line 44: change to "objective identification of PV structures" Lines 47,48, 78, etc.: change to "e.g.," to be consistent with other uses of e.g. Line 66: change to "Bithell et al. (1999)" Line 99: change "that base on" to "that are based on" Line 153: change "south pole" to "South Pole" for consistency Line 175: does "resp" mean "respectively"? Line 208: capitalize "Euclidean" Line 216: change "of a measure" to "of as a measure" Line 229: Do you want to capitalize Seasonal, since Subseasonal is capitalized? Line 286: "Depending on the use case", do you mean "Depending on the case"? Line 380: Do you want "for cutoffs" within parentheses? Line 398: change "independently on" to "independently of" Line 405: change to "Bithell et al. (1999)" We thank the reviewer for these technical and mainly stylistic suggestions. Text changes have been done where the authors deem them appropriate. We refer the reviewer to the uploaded Author's track-changes file for the changes made.